# MSMCE: A novel representation module for classification of raw mass spectrometry data

Fengyi Zhang [1], Boyong Gao[1], Yinchu Wang[2,3,4], Lin Guo[2,3,4], Wei Zhang[2,3,4], Xingchuang Xiong [2,3,4]*

**1** China Jiliang University, College of Information Engineering, Hangzhou, China, **2** National Institute of Metrology, Beijing, China, **3** Key Laboratory of Metrology Digitalization and Digital Metrology for State Market Regulation, State Administration for Market Regulation, Beijing, China, **4** National Metrology Data Center, Beijing, China

* xiongxch@nim.ac.cn

## Abstract

Mass spectrometry (MS) analysis plays a crucial role in the biomedical field; however, the high dimensionality and complexity of MS data pose significant challenges for feature extraction and classification. Deep learning has become a dominant approach in data analysis, and while some deep learning methods have achieved progress in MS classification, their feature representation capabilities remain limited. Most existing methods rely on single-channel representations, which struggle to effectively capture structural information within MS data. To address these limitations, we propose a Multi-Channel Embedding Representation Module (MSMCE), which focuses on modeling inter-channel dependencies to generate multi-channel representations of raw MS data. Additionally, we implement a feature fusion mechanism by concatenating the initial encoded representation with the multi-channel embeddings along the channel dimension, significantly enhancing the classification performance of subsequent models. Experimental results on four public datasets demonstrate that the proposed MSMCE module not only achieves substantial improvements in classification performance but also enhances computational efficiency and training stability, highlighting its effectiveness in raw MS data classification and its potential for robust application across diverse datasets.

## Introduction

Mass spectrometry (MS) is a highly versatile and powerful analytical tool used for detecting, characterizing, and quantifying various analytes based on their observed mass-to-charge ratio (m/z) [1]. With technological advancements, the rate of MS data generation and its complexity have increased significantly. Its characteristics of massive data volume and high dimensionality, coupled with inherent noise and significant signal variability (including peak shifts and intensity fluctuations across samples),

**Data availability statement:** The data underlying the results presented in the study are available from Figshare (https://doi.org/10.6084/m9.figshare.29148629.v1).

**Funding:** This work was supported by the Science & Technology Fundamental Resources Investigation Program (Grant No.2022FY101200) awarded to XX.

**Competing interests:** The authors have declared that no competing interests exist.

pose considerable challenges for data analysis. How to rapidly and accurately interpret these complex MS datasets has become a major challenge in MS data analysis.

In the field of MS data classification, traditional machine learning algorithms, such as Support Vector Machines (SVM), logistic regression, Random Forest, and XGBoost, have been widely applied and often serve as performance benchmarks [2–7]. However, the effective application of these traditional algorithms on MS data typically relies on a series of complex data preprocessing steps. These preprocessing measures aim to address inherent issues in MS signal acquisition, such as peak shifts across different mass spectra, baseline drift, noise interference, and variations in signal intensity, thereby extracting more interpretable biological features to support subsequent biological analysis and enhance the robustness of machine learning models [8]. Therefore, meticulous data preprocessing steps, including denoising, baseline correction, peak detection, and alignment, are usually essential operations. Concurrently, some research has also begun to explore strategies for directly utilizing raw MS data for deep learning classification [9,10], aiming to leverage the ability of deep learning models to automatically learn data representations. Nevertheless, both traditional methods and methods directly utilizing raw MS data need to address the inherent high dimensionality of MS data (a large number of m/z features) and the typically limited sample sizes in specific cohorts [11]. These characteristics can add extra complexity, making it particularly critical to develop robust models without risking overfitting or information loss due to aggressive feature selection. This persistent need for effectively learning discriminative representations from data motivates the exploration of alternative methods that can learn representations more directly from raw MS data.

In recent years, deep learning has emerged as a dominant approach in data analysis, achieving breakthrough advancements in model architectures such as Convolutional Neural Networks (CNNs) (e.g., ResNet [12], DenseNet [13], and EfficientNet [14]), and sequence modeling networks like LSTM [15], and Transformer [16]. Its rapid development has surpassed the limitations of traditional machine learning in many aspects. Consequently, researchers have actively explored its applications in the field of mass spectrometry, encompassing various areas such as disease diagnosis [17,18] and peptide sequencing and identification [19–24]. A key advantage is that deep can capture complex patterns and latent features from raw MS data [24], showcase its significant potential in MS classification. However, a core challenge in MS data analysis remains how to quantitatively and effectively represent MS vectors. To address this challenge, particularly while maintaining high analytical performance at lower computational costs, many deep learning methods focus on embedding high-dimensional MS vectors into lower-dimensional spaces. These embedding methods have varied focuses; some research utilizes graph neural networks to embed molecular structural information for mass spectrometry prediction [25], while other studies have developed advanced embedding methods for tasks such as mass spectrometry clustering and similarity assessment. For example, Spec2Vec [26], inspired by representation learning techniques from natural language processing, learns embedded representations of MS vectors from large-scale MS data, verifying that the relationships between vector fragments can reflect the structural similarity of different compounds. MS2DeepScore

[27] employs a Siamese neural network to learn low-dimensional embeddings of MS vectors that are used for predicting the structural similarity between chemical compounds. GLERMS [28] enhances the low-dimensional representation of MS vectors through contrastive learning, significantly improving compound identification and MS clustering performance.

In the field of image classification, multi-channel images leverage complementary information across different channels can provide a more comprehensive description of the target than single-channel images [29]. In this context, 'channels' typically refer to color channels (e.g., Red, Green, Blue), where each pixel in an image is formed by a combination of values from these color channels. We posit that multi-channel representations can offer analogous benefits for high-dimensional MS data. Compared to single-channel representations, multi-channel representation by fostering inter-channel information correlation, can generate more expressive features. This enables model to learn deeper patterns that are often unrevealed by single-channel representations, thereby enhancing the model's capacity for recognize complex data patterns and improving classification performance. CNNs are exceptionally well-suited for such multi-channel representation learning due to their powerful feature extraction capabilities. CNNs can extract latent features from MS vectors by capturing spatial invariance (such as translation and scale invariance) and encoding these into feature maps, with pooling operations further refining structural features [30]. By stacking convolutional layers and applying nonlinear transformations, CNNs can dynamically generate multi-channel embedding representations, effectively capturing information from different aspects and hierarchical levels within MS data. In our work, 'channel' specifically refers to a distinct feature map produced by convolutional layer. Each such channel embodies a learned embedding representation or a filtered perspective of the input data. This multi-channel embedding representations not only enrich feature diversity but also enhance the model's capability to process high-dimensional data. The inherent ability of CNNs to integrate feature extraction with classification contributes to their robustness, allowing them to maintain high accuracy even when processing MS data corrupted by noise or signal shifts. Studies have demonstrated that CNNs have been widely applied to MS data classification tasks, and have demonstrated excellent performance [31–34].

Therefore, this study aims to design a Multi-Channel Embedding Representation Module that constructs inter-channel dependencies to generate more expressive feature representations, thereby improving both the classification accuracy and generalization ability of models for raw MS data. It is important to emphasize that the data processing methodology adopted in this study differs from traditional feature engineering, which aims to extract interpretable biological features. Out work is an extension of existing research that directly utilizes raw MS data for deep learning classification [9,10], focusing on optimizing the feature representation when raw MS data is input into deep learning models. The main contributions of this study are as follows:

1. Multi-Channel Embedding Representation for Mass Spectrometry Data: By representing one-dimensional MS vector as multi-channel embedded representations, the proposed method enriches the feature expressiveness of the raw data, enabling the model to capture deeper associations between different features during the learning process. Additionally, the dimensionality of the embedded representation output by the MSMCE module is more aligned with the input expectations of convolutional and sequence models.

2. Learnable Representation Module for End-to-End Training Framework: The proposed MSMCE module is a learnable representation module capable of integration with various deep learning classifiers, constructing an end-to-end training framework. This enables the joint optimization of feature representation learning and classification tasks, automatically learning task-specific feature representations within a unified training process to enhance classifier performance.

3. Enhanced Computational Efficiency and Training Stability: While improving classification accuracy, the MSMCE module also contributes to computational efficiency. For CNN architectures, MSMCE reduces computational demands and GPU memory usage through its efficient embedding mechanism. For sequence models, the structured input provided by MSMCE significantly improves training stability when handling high-dimensional MS data, avoiding potential "mode collapse".

## Materials and methods

### Multi-channel embedding module (MSMCE)

The core mechanism of the MSMCE module lies in performing dynamic feature representation learning on input MS vectors. Specifically, it is designed as a preceding feature learning unit for deep learning classification models. It accepts a batch of 1D MS vectors, $X \in R^{B \times D}$, where $B$ represents the batch size and $D$ denotes the dimensionality of each MS vector. Through its internal learnable parameters, MSMCE dynamically transforms each input 1D MS vector (i.e., single-channel) into a more information-rich and expressive multi-channel embedded representation. This process is designed to capture both global contextual information and local fine-grained structural features within the data, thereby providing a more discriminative feature input for downstream classification tasks. Fig 1 illustrates the structural design of the proposed MSMCE module.

**Fully connected encoder.** The input matrix $X$ first passes through a two−layer fully connected network. After the first linear transformation, $X$ undergoes layer normalization, ReLU (Rectified Linear Unit) activation, and Dropout. ReLU is an activation function commonly used in deep learning that introduces non-linearity by outputting the input directly if it is positive and zero otherwise; this helps mitigate the vanishing gradient problem and often improves computational efficiency. Dropout is a regularization technique applied during training where a random proportion of neuron outputs in a layer are temporarily ignored, which helps to prevent the model from overfitting to the training data and thus enhances its generalization ability to unseen data. These components collectively contribute to robust feature learning and model

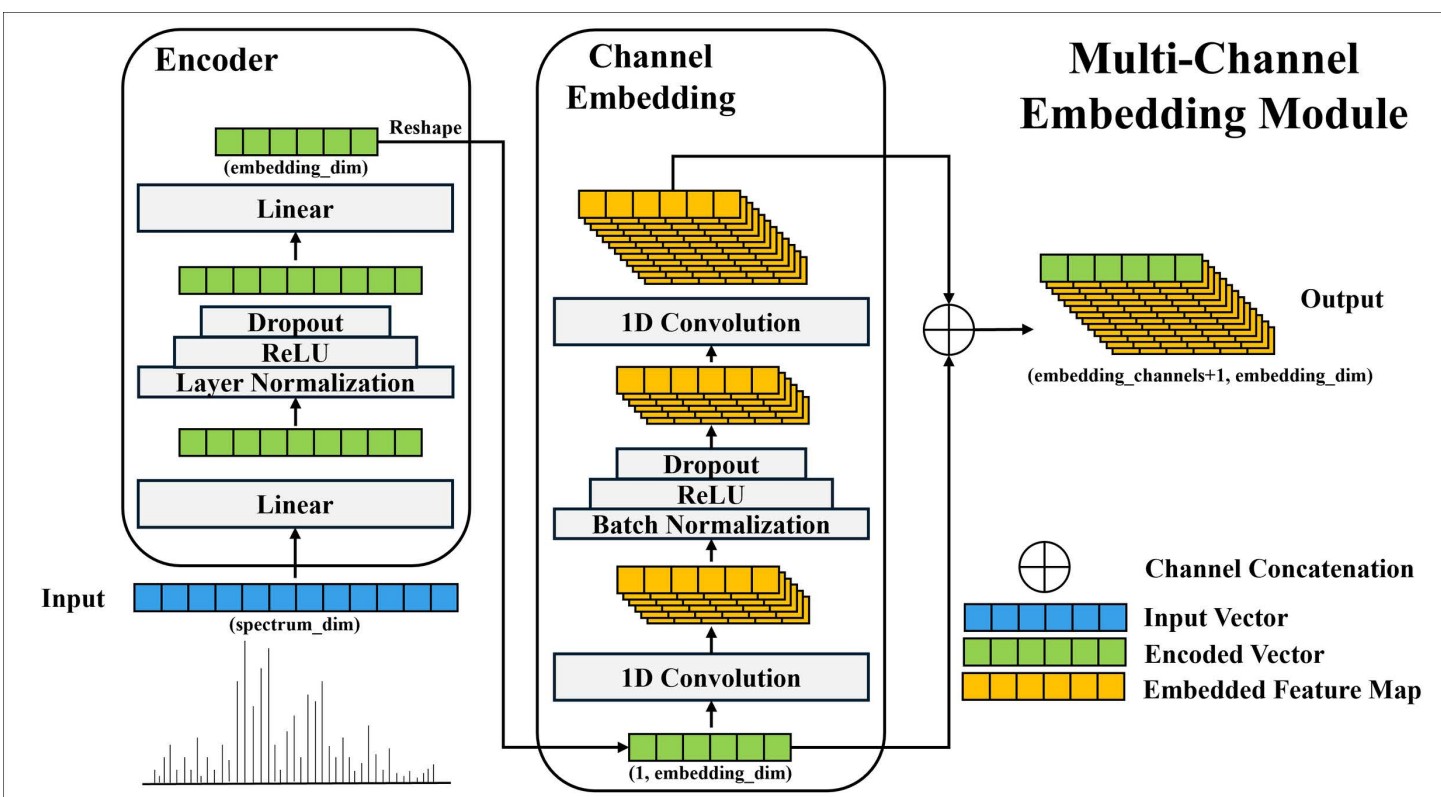

**Fig 1. Structural diagram of the multi-channel embedding (MSMCE) module.** This module consists of three main components: Encoder, Channel Embedding, and Channel Concatenation, which are designed to enhance the feature representation capability of mass spectrometry data. Note: This diagram illustrates the processing of a single MS vector through the module; in practice, the module processes a batch of B such vectors concurrently.

stability. Following these operations, this initial two-layer network effectively compresses the dimensionality of the input vectors and extracts global features to generate an initial encoded vector, providing a more refined feature representation for the subsequent channel embedding module. The mathematical formulation is as follows:

$$H_1 = \text{ReLU}\left(\text{LayerNorm}\left(XW_1 + b_1\right)\right) \tag{1}$$

$$H_1 = \text{Dropout}\left(H_1\right) \tag{2}$$

$$E = H_1 W_2 + b_2 \tag{3}$$

Here, $W_1 \in R^{D \times 2048}$ is the weight matrix of the first fully connected layer, it linearly projects the $D$-dimensional input MS vector $X$ into a $2048$-dimensional intermediate feature space. $b_1 \in R^{2048}$ is the bias vector for this layer, following $XW_1$ linear transformation, it is added to each of the 2048 dimensions of the intermediate result. $W_2 \in R^{2048 \times d}$ is the weight matrix of the second fully connected layer, and $b_2 \in R^d$ is the bias vector for this layer. The resulting latent embedding representation $E \in R^{B \times d}$ is the global embedding representation of the MS data, where $d$ is the embedding dimension.

The fully connected encoder aims to reduce the dimensionality of high-dimensional MS vectors through nonlinear transformations, preserving key features while eliminating redundant information. This results in a more compact feature representation, facilitating more effective downstream processing.

**Channel embedding module.** For the subsequent extraction of local patterns from the global embedding representation $E$, it is reshaped to introduce a channel dimension:

$$E' = E \otimes 1 \in R^{B \times 1 \times d} \tag{4}$$

The reshaped tensor is then processed through two consecutive 1D convolutional layers to further extract local features, ultimately producing multi-channel embedding representation. After the first convolutional layer, batch normalization, ReLU activation, and a Dropout are applied to ensure the model's generalization capability. The second convolutional layer takes the output of the first layer and extracts deeper local features. Both convolutional layers use the same kernel size to maintain consistency between input and output along the length dimension, ensuring feature scale invariance. The mathematical formulation of the Channel Embedding module is as follows:

$$C_1 = \text{ReLU}\left(\text{BatchNorm}\left(\text{Conv1D}\left(E', K_1\right)\right)\right) \tag{5}$$

$$C_2 = \text{Conv1D}\left(\text{Dropout}\left(C_1\right), K_2\right) \tag{6}$$

Here, $K_1$ is the kernel for the first 1D convolutional layer with shape $\left(\frac{C}{2}, 1, 3\right)$, where $out\_channels = \frac{C}{2}$, $in\_channels = 1$, $kernel\_size = 3$. The output $C_1 \in R^{B \times \frac{C}{2} \times d}$ is an intermediate latent embedding from the channel embedding module. $C$ denotes the total number of output channels by this module. $K_2$ is the kernel for the second 1D convolutional layer with shape $\left(C, \frac{C}{2}, 3\right)$, where $out\_channels = C$, $in\_channels = \frac{C}{2}$, $kernel\_size = 3$, and its output $C_2 \in R^{B \times C \times d}$, constitutes the final multi-channel embedding representation, which captures local patterns and fine-grained details from the initial global embedding $E$.

The Channel Embedding module is designed to leverage convolution operations to capture the local patterns and fine-grained details from the input data. It transforms global embedding representation into multi-channel embedding representation, thereby providing a structured format that is dimensionally more aligned with downstream classification tasks.

**Channel concatenation.** To generate the final output tensor, the initial global embedding $E'$ is concatenated with the multi−channel embedding representation $C_2$ along the channel dimension. This concatenation operation allows for the direct integration of the global context captured by $E'$ with the detailed local patterns learned by $C_2$. The resulting fused

embedding representation $O$ thus combines information from both the encoder and the channel embedding module, providing a richer and more comprehensive input for subsequent classification tasks. The channel concatenation can be expressed as:

$$O = \text{Concat}\left(E', C_2\right) \in R^{B \times (1+C) \times d} \tag{7}$$

Here, $O$ represents the final output tensor of the MSMCE module. The fused embedding representation is more diverse and representative, effectively capturing the key feature information within MS data while significantly reducing the dimensionality of the original MS vectors, thereby improving computational efficiency.

The proposed MSMCE module provides an efficient and robust method for representation learning directly from raw MS data. The module's core innovation lies in its ability to first compress high-dimensional, sparse MS vector to extract key global contextual information. Subsequently, using cascaded convolutional layers, it deeply mines this global representation to capture local, fine-grained patterns that are difficult to discern in the raw MS data. By organically combining these two capabilities, global summarization and local detail extraction, MSMCE generates more comprehensive and hierarchical feature representations, thereby greatly enhancing its capacity to characterize complex MS data.

## Dataset description

This study evaluates the proposed method using four publicly available datasets. Table 1 provides detailed information on all datasets, including mass spectrometry instruments used and the number of classified samples after data processing.

1. Canine Sarcoma Dataset [35]: This dataset contains 1 healthy and 11 sarcoma histology types. It can be formulated as either a binary classification task or a 12-class classification task. This study analyzes data acquired in positive ion mode.

2. NSCLC Dataset [36]: This dataset includes the two major histological subtypes of non-small cell lung cancer (NSCLC), namely adenocarcinoma (ADC) and squamous cell carcinoma (SCC), and is utilized herein for a binary classification task.

**Table 1. Dataset information.**

| Dataset | Instrument | Classes | Files | Mass range |
|---|---|---|---|---|
| Canine sarcoma | Synapt G2-S Q-TOF | Healthy | 40 | 100–1600 |
| | | Myxosarcoma | 5 | |
| | | Fibrosarcoma | 30 | |
| | | Hemangiopericytoma | 10 | |
| | | Malignant peripheral nerve tumor | 5 | |
| | | Osteosarcoma | 25 | |
| | | Undifferentiated pleomorphic | 25 | |
| | | Rhabdomyosarcoma | 5 | |
| | | Splenic fibrohistiocytic nodules | 5 | |
| | | Histiocytic sarcoma | 5 | |
| | | Soft tissue sarcoma | 5 | |
| | | Gastrointestinal stromal sarcoma | 5 | |
| NSCLC | LTQ Orbitrap Elite | ADC | 6 | 400–1600 |
| | | SCC | 6 | |
| CRLM | Orbitrap Fusion Lumos | Control | 30 | 400–1600 |
| | | CRLM | 30 | |
| RCC | Q Exactive HF | Control | 174 | 70–1060 |
| | | RCC | 82 | |

3. CRLM Dataset [37]: This dataset consists of colorectal liver metastases (CRLM) tissues and normal liver tissues, serving as a binary classification task.

4. RCC Dataset [38]: This dataset includes samples from renal cell carcinoma (RCC) patients subjects and healthy control subjects, formulated as a binary classification task. This study analyzes data acquired in positive ion mode.

## Data processing workflow

Raw mass spectrometry data, fundamentally, allows a single mass spectrum sample to be conceptualized as a sequence of m/z and corresponding intensity value pairs: $S = [((m/z)_1, I_1), ((m/z)_2, I_2), \ldots, ((m/z)_n, I_n)]$. During MS acquisition, the data evolves with retention time (RT), meaning each mass spectrum sample is associated with a specific time point $t_i$. Consequently, a complete MS file can be viewed as a series of mass spectrum samples over time, $T = [(S_1, t_1), (S_2, t_2), \ldots, (S_n, t_n)]$, where each $S_j$ is itself a high-dimensional intensity vector across numerous m/z values. This inherent complexity, with variations along both the m/z and RT dimensions, necessitates structured processing to generate a consistent input format for deep learning models.

The data processing workflow in this study is based on related work that directly utilizes raw mass spectrometry data for classification. The overall pipeline, illustrating the complete process from the input of mass spectrometry file to the generation of feature matrix, is depicted in Fig 2. For LC-MS datasets, including NSCLC, CRLM, and RCC, we referenced the data processing method described in [9]. Raw mass spectra read from the MS files were first binned along the m/z axis using a bin width of 0.1 Da, thereby generating a fixed number of m/z features for each dataset. For the RT dimension, mass spectra were aggregated within 10-second binning windows. Specifically, within each 10-second binning window, the intensity values for each corresponding m/z dimension from all mass spectra were summed and then averaged to create a representative mass spectrum sample for that RT binning window. For SpiderMass datasets, such as the Canine Sarcoma dataset, the data processing workflow reference method outlined in [10]. Initially, mass spectra with a total ion count (TIC) below $1 \times 10^4$ were filtered out to ensure data quality. Subsequently, the retained mass spectra were also binned along the m/z axis with a bin width of 0.1 Da to achieve a uniform data dimension, ensuring comparability between different samples. In summary, whether it involves m/z binning and RT window aggregation for LC-MS datasets, or TIC filtering and m/z binning for SpiderMass datasets, the processing ultimately creates a two-dimensional feature matrix for each mass spectrometry file. In this matrix, rows correspond to the mass spectrum samples that have undergone RT window aggregation or filtering and retention, while columns correspond to the m/z feature dimension. Each row in the two-dimensional feature matrix (i.e., a processed mass spectrum instance) is then treated as an independent input sample for the training of subsequent deep learning models. Fig 3 displays the t-SNE dimensionality reduction visualizations of the samples from all datasets after the data processing workflow, intuitively revealing the distinct intrinsic structures and class distribution characteristics among the different datasets.

## Training strategy

For each dataset, we first partition the file list of the raw MS data. Specifically, stratified sampling is performed based on the class labels associated with each file, dividing the list into a training set (90%) and a fixed hold-out test set (10%), thereby minimizing potential biases in subsequent model training and evaluation due to class distribution imbalances. Subsequently, all files within both the training and test sets were independently subjected to the data processing workflow. During this process, each file was transformed into multiple representative mass spectrum instances. All feature rows derived from the training set were then stacked to form the final training set feature matrix; similarly, all feature rows

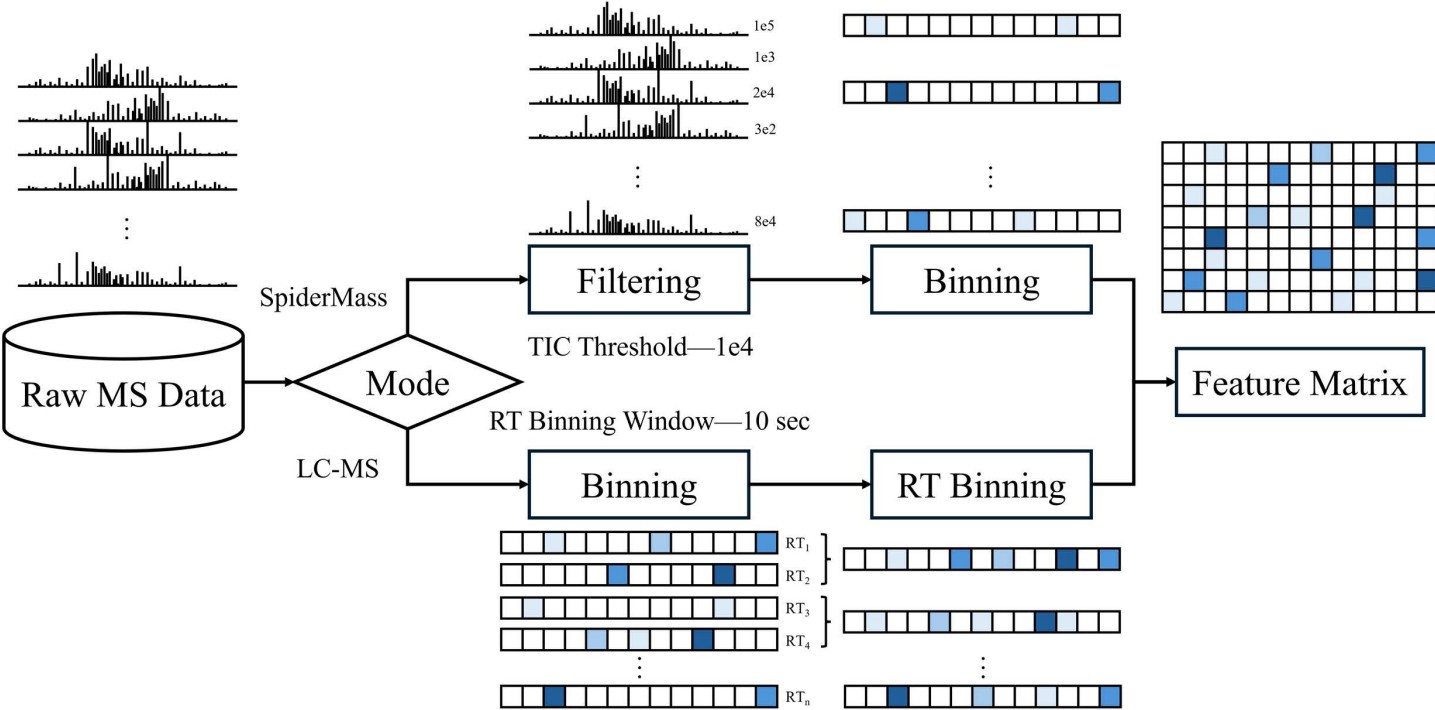

**Fig 2. Mass spectrometry data processing workflow.** This figure illustrates the preprocessing pipeline for mass spectrometry data, covering both SpiderMass and LC-MS data processing methods. The final output is a Feature Matrix, which serves as the input for subsequent analyses.

from the test set were used to construct the test set feature matrix in the same manner. Finally, both of these resulting feature matrices, prior to model input, were subjected to TIC normalization to reduce the data's dynamic range and improve feature comparability across samples. The training set was then used for K-fold stratified cross-validation, with K set to 6. For each of the K folds, the data was further divided into a training fold ($\frac{K-1}{K}$ of the training set) and a validation fold ($\frac{1}{K}$ of the training set), with a random seed set to ensure that all models evaluated on the same dataset used identical fold splits.

For all deep learning experiments, the MSMCE module was configured with 256 embedding channels and an embedding dimension of 1024. Models were trained using the Adam optimizer with an initial learning rate of $1 \times 10^{-3}$ and a weight decay (L2 regularization) coefficient of $1 \times 10^{-5}$ to mitigate overfitting. To address potential class imbalance within each training fold, class weights were computed and applied to the cross-entropy loss function. These weights were automatically set based on the sample count for each class, specifically, classes with fewer samples were assigned higher weights, while classes with more samples received lower weights. This strategy aimed to balance the contribution of different classes to the loss function during training, thereby alleviating the impact of class imbalance on the final classification results. Additionally, a learning rate scheduler (ReduceLROnPlateau) was configured to automatically adjust the learning rate when validation fold performance stagnated. Specifically, if performance did not improve for 5 epochs, the learning rate was decayed by a factor of 0.1, thus improving training stability and model convergence efficiency. An early stopping mechanism based on validation loss, with a patience of 10 epochs, was employed to prevent further overfitting and to dynamically preserve the model weights that achieved the best performance on the validation fold.

All deep learning models were trained for a maximum of 64 epochs, unless halted earlier by an early stopping mechanism. For machine learning models, standard implementations with default parameter configurations were used for

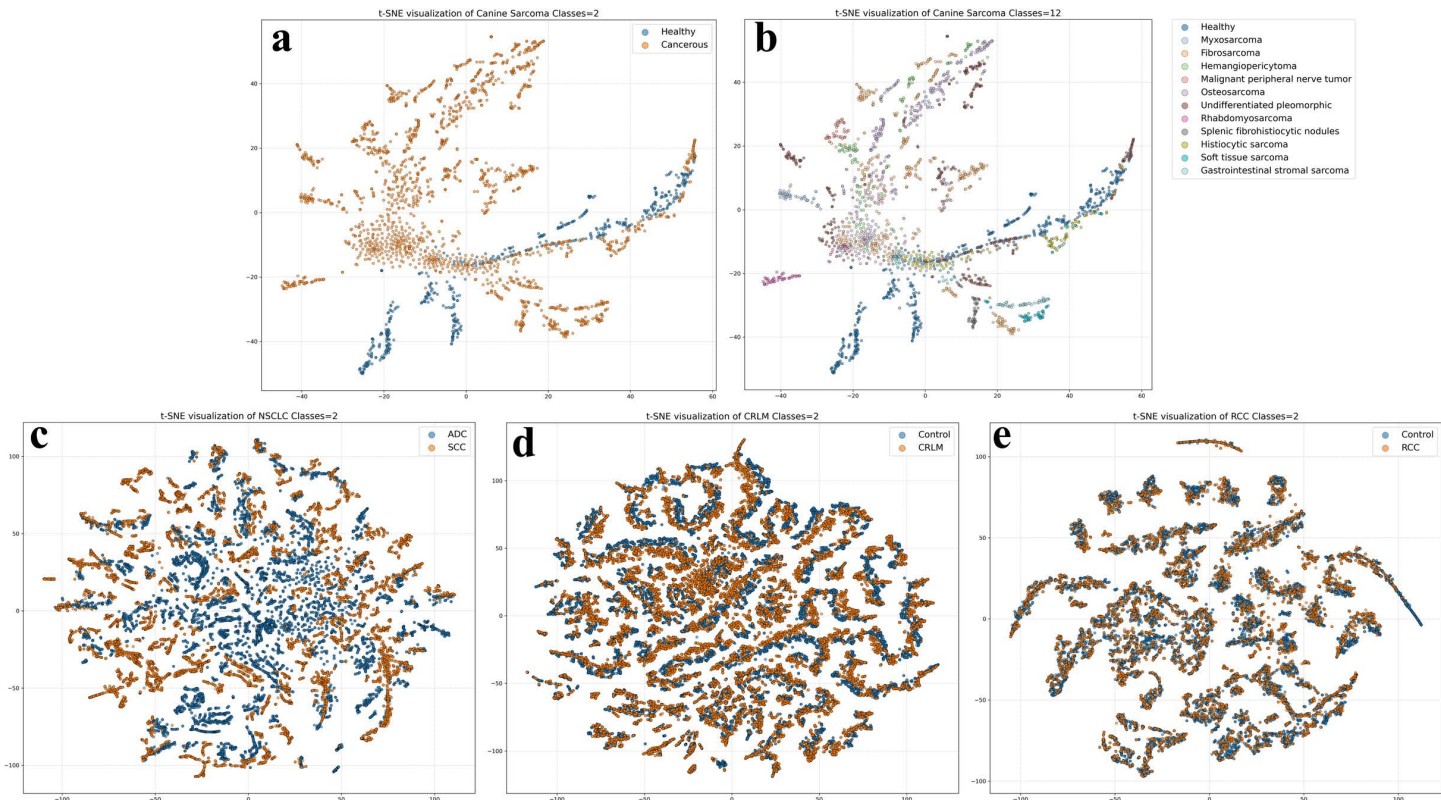

**Fig 3. t-SNE visualization of all studied datasets.** This figure shows the two-dimensional t-SNE visualizations for the four datasets used in this study, illustrating their intrinsic data structures and class separability. (a) Canine Sarcoma dataset shown as a binary classification view; (b) Canine Sarcoma dataset shown as a 12-class view with all sarcoma subtypes; (c) NSCLC dataset; (d) CRLM dataset; (e) RCC dataset.

baseline comparison. Finally, the performance of each model was evaluated on the independent and fixed hold-out test set, and statistical tests were conducted based on the corresponding performance metrics. All stochastic processes, including data partitioning and k-fold generation, were controlled by a unified global random seed to ensure the reproducibility of the experiments.

## Training process

To validate the effectiveness of MSMCE in feature representation, we integrate the proposed module with various mainstream deep learning architectures, including CNNs such as ResNet, EfficientNet, and DenseNet, as well as sequence modeling networks like LSTM and Transformer. These models have been widely applied across different tasks, each possessing unique structural characteristics and feature extraction capabilities. It is important to emphasize that this study does not adopt an ensemble learning strategy. Instead, the MSMCE module serves as a feature representation layer that is directly connected to classification models at the structural level, optimizing feature input for improved classification performance.

To ensure compatibility of the MSMCE module with various types of classification models, we adapted the input layers of each model accordingly. For CNN models, we modified the number of input channels in their initial convolutional layer to match the number of channels in the multi-channel embedding representation output by the MSMCE module. This ensured that the generated embedding representation could be directly and effectively processed by standard

convolutional operations. For sequence models, in their baseline configuration, the model received the single-channel MS vector and segmented it to form a sequence input. Specifically, the LSTM model processed this segmented sequence vector through its recurrent layered architecture, extracting features from the sequence's final state for classification. The transformer model, on the other hand, first created an embedding representation for the segmented sequence, prepended a classification token (CLS Token) to the head of the sequence, and then utilized its self-attention mechanism to learn contextual representations of the sequence, ultimately performing classification based on this classification token. When integrated with MSMCE, both sequence models directly used the MSMCE's output as their input sequence, where the embedded channels were treated as the sequence length, and the embedding dimension corresponded to the feature length of each element in the sequence.

As shown in Fig 4, the multi-channel features generated by the MSMCE module are directly fed into the classification model, forming an end-to-end training process. During model training, the MSMCE module and the classification model are jointly optimized, with parameter gradients updated simultaneously under a unified loss function. This approach does not rely on manual feature engineering but instead leverages a deep embedding mechanism to automatically extract and organize feature information from the data, enabling a learnable, data-driven representation of the input. The objective is to ensure that different components of the model work collaboratively, maximizing the synergy between the representation module and the classification model, ultimately achieving efficient modeling and classification of raw MS data.

## Results

To evaluate the effectiveness of the proposed MSMCE module, we conducted a systematic comparison between models incorporated with the MSMCE module and their original counterparts without MSMCE across four different datasets. For all experiments, the batch size for the NSCLC, CRLM, and RCC datasets was set to 64, while the batch size for the Canine Sarcoma dataset was set to 32. For all multi-class classification tasks reported in this paper, the overall Accuracy and F1-Score metrics were obtained by first calculating these metrics for each class independently and then taking their unweighted average ('macro' averaging).

To assess the statistical significance of performance improvements, we employed the paired Wilcoxon signed-rank test to compare models integrated with MSMCE against their respective baselines. The paired samples for each test consisted of the performance scores from the 6 individual folds of the cross-validation (i.e., $N = 6$ for each test). All p-values were calculated from a two-sided test, and a value of less than 0.05 was considered statistically significant.

### Large-scale binary classification datasets

The NSCLC, CRLM, and RCC datasets have large sample sizes and simple class structures, as they are all binary classification tasks. Furthermore, these datasets exhibit relatively balanced class distribution, providing an ideal testing

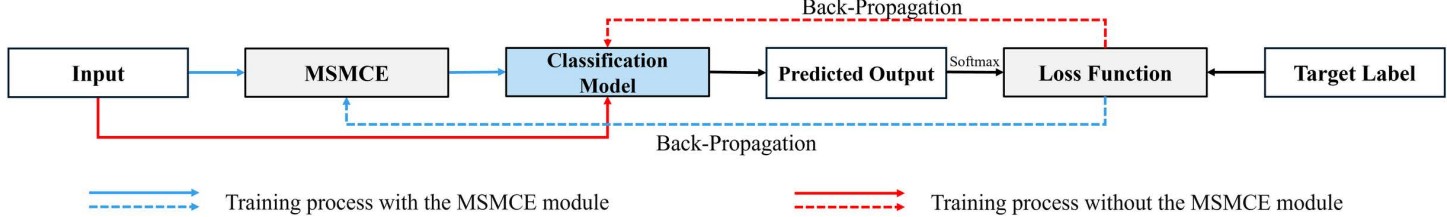

**Fig 4. Comparison between the training process with MSMCE and without MSMCE.** In the blue path, the input data first passes through the MSMCE module for feature transformation before being fed into the classification model for training. The loss gradient optimizes both the classification model and the MSMCE module. In the red path, the input data is directly fed into the classification model, and the loss gradient is only used to optimize the classification model.

environment for evaluating the feature representation learning capability of the proposed MSMCE module. Conducting experiments on these large-scale datasets aims to validate the effectiveness and robustness of the MSMCE approach when dealing with high-volume data, particularly in assessing its comprehensive performance in extracting discriminative features, improving classification accuracy, and handling balanced data distributions effectively.

The experimental results demonstrate that deep learning models integrated with MSMCE module achieved significant improvements in classification performance across all three datasets compared to their original counterparts. As shown in Table 2, on the NSCLC dataset, MSMCE-ResNet50 achieved an accuracy of 0.9785, a statistically significant increase of 1.28% (p = 0.03, N = 6) compared to the original model accuracy of 0.9661. For the EfficientNetB0 model, the high p-value (p = 0.84) suggests that the baseline model's performance was already approaching a "performance ceiling" on this dataset, leaving little room for statistically significant improvement from the MSMCE module. This is further corroborated by the relatively simple intrinsic structure of the dataset, as revealed by its t-SNE visualization in Fig 3(c), where the different classes already exhibit clear separability. As detailed in Table 3, on the CRLM dataset, the accuracy of MSMCE-DenseNet121 improved by 6.66% (p = 0.03, N = 6) relative to DenseNet121. Furthermore, as presented in Table 4, on the RCC dataset, the F1-Score for the DenseNet121 increased from 0.6946 to 0.7952, an improvement of 14.48% (p = 0.03, N = 6). For the EfficientNetB0 and LSTM models, the non-significant p-values may reflect a mismatch between their specific architectures and the intrinsic characteristics of the data. When a model's inductive bias (e.g., spatial locality for CNNs or sequential dependency for LSTMs) is not key to effective discrimination for this dataset, the performance gains from MSMCE's enhanced representation can be limited by the classifier itself, thus not reaching statistical significance.

Moreover, across these three datasets, the Transformer model, after being integrated with the MSMCE module, successfully avoide the extremely low Accuracy and F1-Score observed in the baseline. It was able to converge stably and the learn effective discriminative features, which further underscores the significant role of the MSMCE module in optimizing input representations and improving the training dynamics of complex models. To more intuitively illustrate this critical transformation brought by the MSMCE module, Fig 5 shows a comparison of the confusion matrices for the Baseline Transformer and the MSMCE-Transformer on the NSCLC, CRLM, and RCC datasets. The figure clearly reveals how the MSMCE module helps the Transformer escape the predicament of predicting all samples as a single class and achieve effective discrimination between the two classes.

It is noteworthy that on the NSCLC dataset, the RF model achieved the highest F1-Score of 0.9935 among all tested models. However, on the CRLM and RCC datasets, the RF model's performance did not surpass that of the CNN models integrated with MSMCE. This may suggest that the inherent feature space of the NSCLC dataset is particularly well-suited for learning by tree-based ensemble methods like Random Forest. Nevertheless, the key finding is that the MSMCE

**Table 2. Comparison of classification performance of different models on the NSCLC dataset.**

| NSCLC | Baseline | | MSMCE | |
|---|---|---|---|---|
| Classes = 2 | Accuracy | F1-Score | Accuracy | F1-Score |
| RF | 0.9935 ± 0.00 | 0.9935 ± 0.00 | – | – |
| SVM | 0.9690 ± 0.00 | 0.9690 ± 0.00 | – | – |
| LDA | 0.8879 ± 0.01 | 0.8879 ± 0.01 | – | – |
| ResNet50 | 0.9661 ± 0.00 | 0.9661 ± 0.00 | 0.9785 ± 0.00 (p = 0.03) | 0.9785 ± 0.00 (p = 0.03) |
| DenseNet121 | 0.9659 ± 0.00 | 0.9659 ± 0.00 | 0.9806 ± 0.00 (p = 0.03) | 0.9806 ± 0.00 (p = 0.03) |
| EfficientNetB0 | 0.9769 ± 0.00 | 0.9769 ± 0.00 | 0.9782 ± 0.00 (p = 0.84) | 0.9782 ± 0.00 (p = 0.84) |
| LSTM | 0.7342 ± 0.26 | 0.6508 ± 0.35 | 0.9798 ± 0.00 (p = 0.03) | 0.9798 ± 0.00 (p = 0.03) |
| Transformer | 0.4999 ± 0.00 | 0.3333 ± 0.00 | 0.9813 ± 0.00 (p = 0.03) | 0.9813 ± 0.00 (p = 0.03) |

Baseline column represents the models without the MSMCE module integrated, while the MSMCE column denotes the corresponding models with the MSMCE module integrated.

**Table 3. Comparison of classification performance of different models on the CRLM dataset.**

| CRLM | Baseline | | MSMCE | |
|---|---|---|---|---|
| Classes = 2 | Accuracy | F1-Score | Accuracy | F1-Score |
| RF | 0.9087 ± 0.00 | 0.9085 ± 0.00 | – | – |
| SVM | 0.8939 ± 0.00 | 0.8938 ± 0.00 | – | – |
| LDA | 0.9026 ± 0.00 | 0.9026 ± 0.00 | – | – |
| ResNet50 | 0.8818 ± 0.02 | 0.8816 ± 0.02 | 0.9212 ± 0.01 (p = 0.03) | 0.9212 ± 0.01 (p = 0.03) |
| DenseNet121 | 0.8643 ± 0.03 | 0.8641 ± 0.03 | 0.9219 ± 0.01 (p = 0.03) | 0.9219 ± 0.01 (p = 0.03) |
| EfficientNetB0 | 0.8870 ± 0.00 | 0.8867 ± 0.00 | 0.9189 ± 0.00 (p = 0.03) | 0.9188 ± 0.00 (p = 0.03) |
| LSTM | 0.5000 ± 0.00 | 0.3333 ± 0.00 | 0.9221 ± 0.01 (p = 0.03) | 0.9220 ± 0.01 (p = 0.03) |
| Transformer | 0.5000 ± 0.00 | 0.3333 ± 0.00 | 0.9154 ± 0.01 (p = 0.03) | 0.9153 ± 0.01 (p = 0.03) |

**Table 4. Comparison of classification performance of different models on the RCC dataset.**

| RCC | Baseline | | MSMCE | |
|---|---|---|---|---|
| Classes = 2 | Accuracy | F1-Score | Accuracy | F1-Score |
| RF | 0.8055 ± 0.00 | 0.7369 ± 0.01 | – | – |
| SVM | 0.6754 ± 0.00 | 0.4811 ± 0.00 | – | – |
| LDA | 0.6352 ± 0.01 | 0.6056 ± 0.02 | – | – |
| ResNet50 | 0.7803 ± 0.01 | 0.7431 ± 0.02 | 0.8082 ± 0.01 (p = 0.03) | 0.7872 ± 0.01 (p = 0.03) |
| DenseNet121 | 0.7494 ± 0.01 | 0.6946 ± 0.02 | 0.8156 ± 0.01 (p = 0.03) | 0.7952 ± 0.00 (p = 0.03) |
| EfficientNetB0 | 0.8109 ± 0.01 | 0.7826 ± 0.01 | 0.8064 ± 0.01 (p = 0.69) | 0.7864 ± 0.01 (p = 0.69) |
| LSTM | 0.6991 ± 0.05 | 0.5354 ± 0.16 | 0.7865 ± 0.06 (p = 0.16) | 0.7232 ± 0.16 (p = 0.16) |
| Transformer | 0.6777 ± 0.00 | 0.4000 ± 0.00 | 0.7709 ± 0.06 (p = 0.04) | 0.6946 ± 0.16 (p = 0.04) |

module consistently delivered performance enhancements across all tested deep learning models and datasets, highlighting its general utility in optimizing raw MS data representation for improved deep learning classification efficacy.

### Small-scale multi-class classification dataset

To further validate the effectiveness of the MSMCE module, experiments were conducted on the Canine Sarcoma dataset. This dataset is characterized by numerous classes and a limited sample size, and it was utilized for evaluating performance on both binary and 12-class classification tasks.

The experimental results indicate that even on datasets with a limited sample size and numerous classes, models integrated with the MSMCE module still achieve significant performance improvements. As shown in Table 5, for the binary classification task on the Canine Sarcoma dataset, the accuracy of Transformer improved from 0.7848 to 0.9933, an increase of 26.57% (p = 0.03, N = 6). For the 12-class classification task on the Canine Sarcoma dataset, the result in Table 6 show that the accuracy of ResNet50 improved from 0.7235 to 0.9043, an increase of 24.99% (p = 0.03, N = 6). The t-SNE visualization of the 12-class Canine Sarcoma dataset, as show in Fig 3(b), reveals significant class overlaps among its sarcoma subtypes, highlighting the inherent substantial difficulty in distinguishing them. The significant progress made by MSMCE-ResNet50 underscores the effectiveness of the MSMCE module in learning more discriminative representations capable of disentangling these closely related subtypes, even with limited samples per class. Furthermore, the relatively suboptimal performance of the MSMCE-Transformer model on the 12-class Canine Sarcoma task might be attributed to the inherent complexity of the Transformer model in conjunction with the limited number of training samples available for each fine-grained subtype in this multi-class scenario, which may have hindered the model from adequately learning robust discriminative features among all classes.

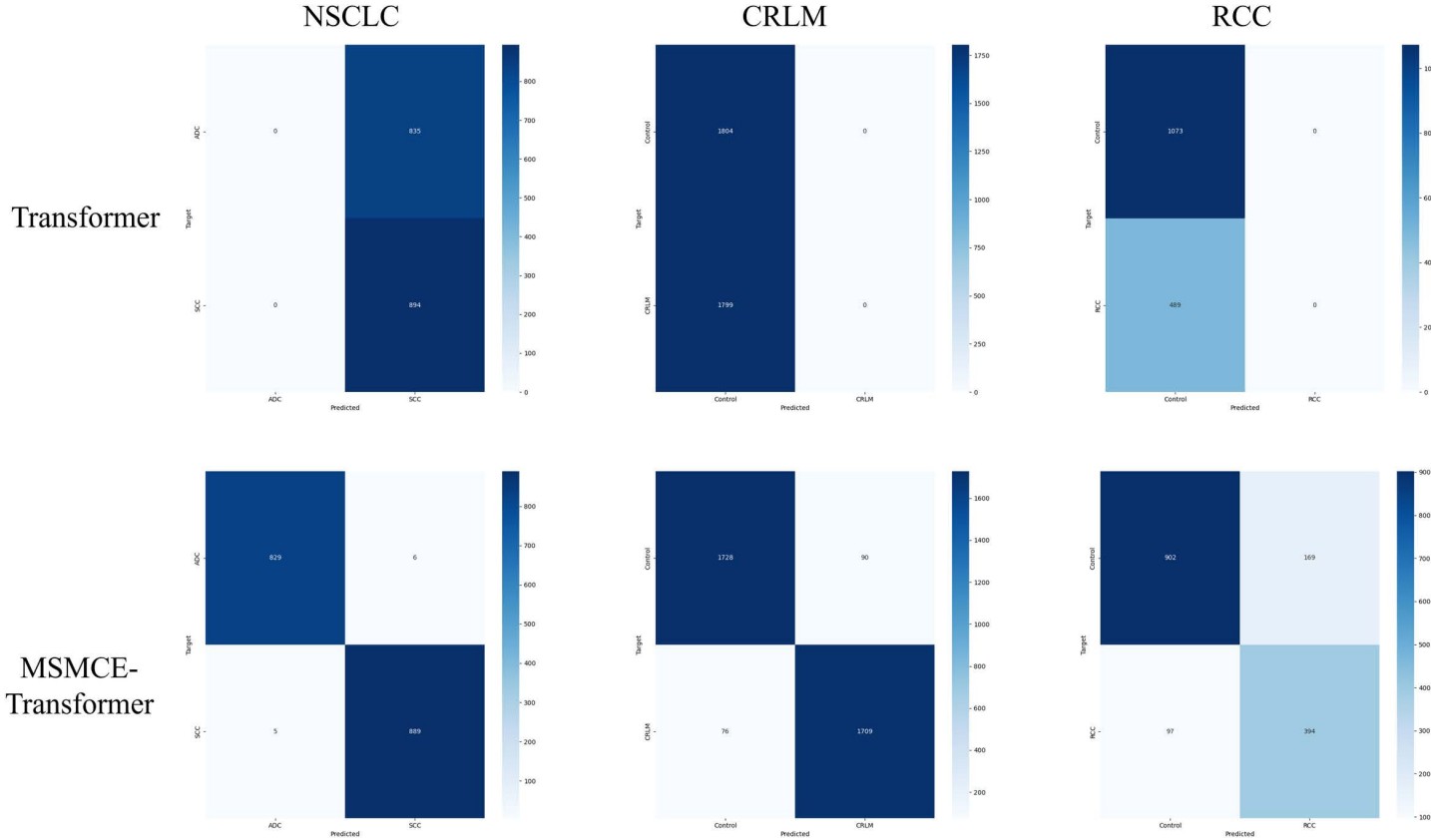

**Fig 5. Comparison of confusion matrices for transformer and MSMCE-transformer on three binary classification datasets.** This figure presents a comparative analysis of the confusion matrices for the baseline Transformer model (top row) and the MSMCE-Transformer model (bottom row) across the NSCLC, CRLM, and RCC datasets. The confusion matrices for the baseline Transformer clearly reveal that the model predicts all samples as a single class, resulting in extremely poor discriminative performance. In stark contrast, the confusion matrices for the MSMCE-Transformer model show a high concentration of values along the diagonal, indicating high numbers of true positives and true negatives.

**Table 5. Comparison of classification performance of different models on the canine sarcoma dataset (Classes = 2).**

| Canine sarcoma | Baseline | | MSMCE | |
|---|---|---|---|---|
| Classes = 2 | Accuracy | F1-Score | Accuracy | F1-Score |
| RF | 0.9836 ± 0.00 | 0.9750 ± 0.00 | – | – |
| SVM | 0.9768 ± 0.00 | 0.9646 ± 0.00 | – | – |
| LDA | 0.9357 ± 0.01 | 0.9082 ± 0.01 | – | – |
| ResNet50 | 0.9671 ± 0.01 | 0.9509 ± 0.02 | 0.9955 ± 0.00 (p = 0.03) | 0.9934 ± 0.00 (p = 0.03) |
| DenseNet121 | 0.9783 ± 0.01 | 0.9679 ± 0.01 | 0.9940 ± 0.00 (p = 0.03) | 0.9912 ± 0.00 (p = 0.03) |
| EfficientNetB0 | 0.9783 ± 0.00 | 0.9683 ± 0.00 | 0.9933 ± 0.00 (p = 0.03) | 0.9901 ± 0.00 (p = 0.03) |
| LSTM | 0.6951 ± 0.24 | 0.4278 ± 0.15 | 0.9955 ± 0.00 (p = 0.03) | 0.9934 ± 0.00 (p = 0.03) |
| Transformer | 0.7848 ± 0.00 | 0.4397 ± 0.00 | 0.9933 ± 0.01 (p = 0.03) | 0.9902 ± 0.01 (p = 0.03) |

**Table 6. Comparison of classification performance of different models on the canine sarcoma dataset (Classes = 12).**

| Canine sarcoma | Baseline | | MSMCE | |
|---|---|---|---|---|
| Classes = 12 | Accuracy | F1-Score | Accuracy | F1-Score |
| RF | 0.8610 ± 0.01 | 0.8427 ± 0.02 | – | – |
| SVM | 0.8498 ± 0.01 | 0.8303 ± 0.01 | – | – |
| LDA | 0.7721 ± 0.02 | 0.7700 ± 0.02 | – | – |
| ResNet50 | 0.7235 ± 0.04 | 0.7224 ± 0.04 | 0.9043 ± 0.03 (p = 0.03) | 0.9244 ± 0.03 (p = 0.03) |
| DenseNet121 | 0.7265 ± 0.04 | 0.7264 ± 0.04 | 0.9178 ± 0.03 (p = 0.03) | 0.9362 ± 0.02 (p = 0.03) |
| EfficientNetB0 | 0.8520 ± 0.02 | 0.8474 ± 0.02 | 0.8969 ± 0.03 (p = 0.03) | 0.9163 ± 0.03 (p = 0.03) |
| LSTM | 0.1854 ± 0.11 | 0.0353 ± 0.04 | 0.8767 ± 0.03 (p = 0.03) | 0.8973 ± 0.03 (p = 0.03) |
| Transformer | 0.1883 ± 0.03 | 0.0263 ± 0.00 | 0.7324 ± 0.10 (p = 0.03) | 0.7352 ± 0.12 (p = 0.03) |

These results suggest that the MSMCE module, by constructing multi-channel dependencies, enhances the expressive power of raw MS data. Even on datasets with limited samples and numerous classes, models integrated with MSMCE maintain high classification performance. This enhanced feature representation improves the model's robustness when handling highly complex and diverse mass spectrometry data.

## Ablation study

To evaluate the actual contribution of the MSMCE module to model performance, we conducted an ablation study on the 12-class Canine Sarcoma dataset. ResNet50 was selected as the base model, and we progressively introduced Encoder, Channel Embedding, and Channel Concatenation. For this ablation study, model variants were trained on the entire training set (not subjected to k-fold splitting) and their performance was subsequently evaluated on the independent hold-out test set, to systematically assess their impact on classification performance.

As shown in Table 7, introducing the Encoder module into the Baseline model leads to a moderate improvement in accuracy and F1-Score, indicating that the Encoder plays a role in initial feature extraction. However, the validation accuracy curve for the Encoder exhibits significant fluctuations (see Fig 6), suggesting that the model's generalization ability is still limited. With the addition of the Channel Embedding module, the model's performance improves significantly, achieving an accuracy of 0.8834 and an F1-Score of 0.8888. This result demonstrates that the Channel Embedding module effectively captures complex relationships between multiple channels, enhancing feature representation. Additionally, the validation performance becomes more stable. Building on this, incorporating the Channel Concatenation, which fuses encoded features with channel-embedded features, further optimizes model performance, reaching its best results. This highlights the importance of feature fusion in enhancing classification performance by integrating global and local feature information.

## Computational efficiency analysis

The computational efficiency of the different deep learning models was evaluated based on two key metrics: Floating Point Operations (FLOPs) and the estimated peak GPU memory footprint during training. For each deep learning architecture,

**Table 7. Ablation study results: Impact of different module combinations on ResNet-50 classification performance.**

| Baseline (ResNet-50) | Encoder | Channel Embedding | Channel Concatenation | Accuracy | F1-Score |
|---|---|---|---|---|---|
| ✓ | – | – | – | 0.7803 | 0.7798 |
| ✓ | ✓ | – | – | 0.8072 | 0.7955 |
| ✓ | ✓ | ✓ | – | 0.8834 | 0.8888 |
| ✓ | ✓ | ✓ | ✓ | 0.9417 | 0.9410 |

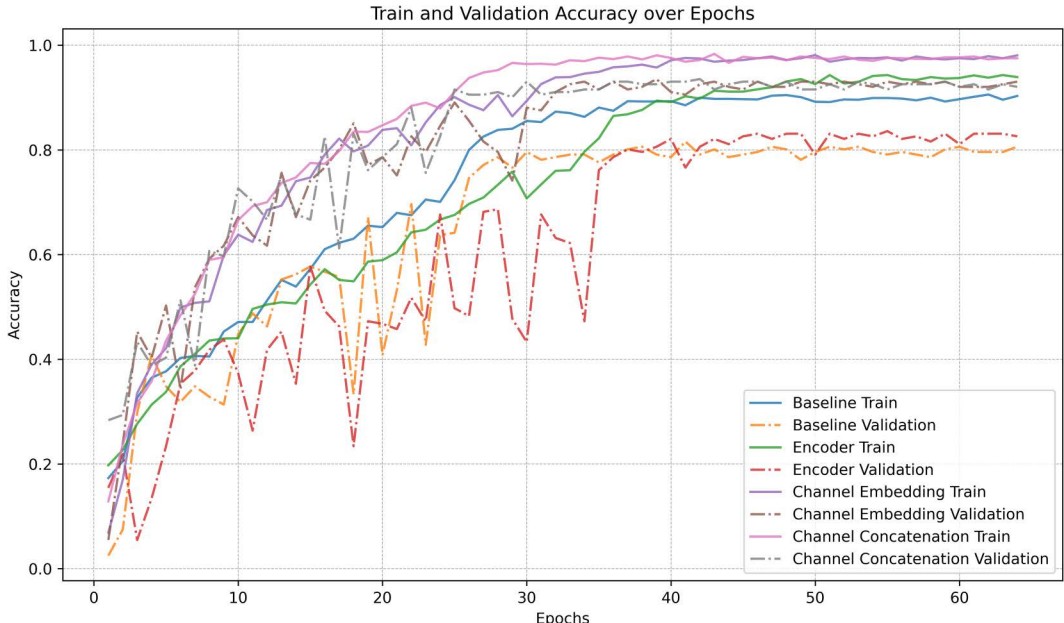

**Fig 6. Accuracy trends during training and validation in the ablation study.** This figure illustrates the accuracy changes on the training and validation sets for ResNet-50 and its variants with progressively introduced MSMCE submodules. The results highlight the impact of different components on model performance. Before introducing the Channel Embedding module, the accuracy curves exhibit significant fluctuations. However, after incorporating Channel Embedding and Channel Concatenation, the accuracy curves become notably more stable, indicating improved model robustness and convergence.

FLOPs were calculated using the profile function from the Python library thop. This calculation was performed by passing a single-sample input tensor of shape (1, spectrum dim) to the function. The model size is the Estimated Total Size (MB), which was estimated to use the summary function from the Python library torchinfo. This estimation was based on an input size of (batch size, spectrum dim), to reflect the peak GPU memory usage during one complete forward and backward pass. As shown in Fig 7, ResNet50, DenseNet121, and EfficientNetB0 exhibit higher FLOPs and larger model sizes, indicating greater computational overhead. However, after being integrated with the MSMCE module, the FLOPs of all these models are substantially reduced (as detailed in S1 File), and the model sizes also decrease, while classification accuracy improves considerably. These findings suggest that the MSMCE module optimizes computational resource utilization by employing an efficient feature embedding approach, enabling models to achieve superior performance with reduced computational cost.

This performance improvement is attributed to the MSMCE module, which first reduces the dimensionality of high-dimensional data before channel embedding. This significantly decreases the data dimension, ensuring that subsequent convolution operations require less computational effort to capture structural information, thereby reducing unnecessary computational overhead in high-dimensional sparse spaces. Additionally, although LSTM and Transformer models experience an increase in computational demand (higher FLOPs and model size) after integrated with MSMCE, their training stability improves significantly. This is a critical trade-off, as it prevents the convergence failures—such as converging to a single-class prediction—observed in the baseline models.

Notably, while the MSMCE module itself increases the model's depth, it ultimately enhances computational efficiency. Furthermore, we observe consistent performance improvements across models of different depths, indicating that the benefits of MSMCE can be effectively integrated with various model architectures, making it suitable for both residual and non-residual network structures.

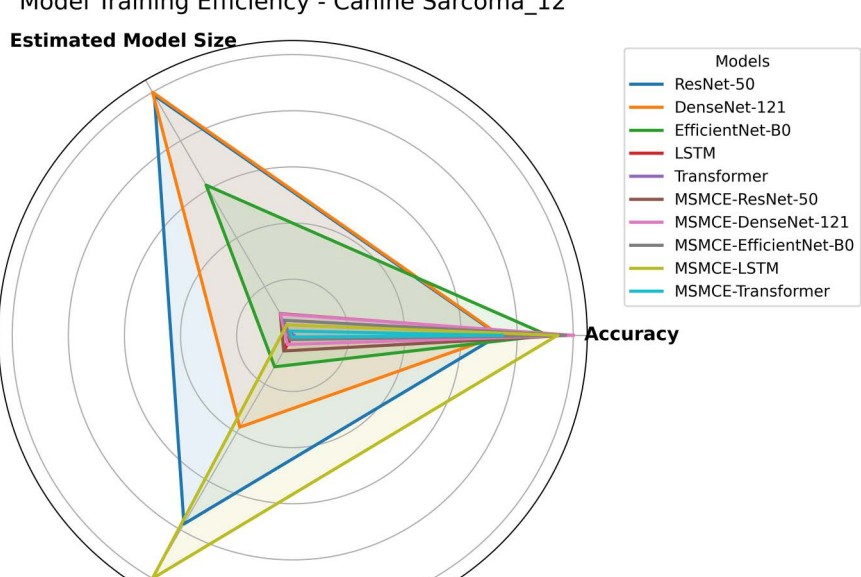

**Fig 7. Radar chart of model training efficiency on the Canine Sarcoma (12-class) Dataset.** This figure presents the computational efficiency of ResNet-50, DenseNet-121, EfficientNet-B0, LSTM, and Transformer, along with their corresponding MSMCE-enhanced versions. It can be observed that MSMCE-enhanced models achieve significantly improved classification accuracy while maintaining lower computational costs (FLOPs) and smaller model sizes. Radar charts for training efficiency on other datasets are provided in the supplementary materials.

## Discussion

The performance of machine learning methods is heavily dependent on the choice of data representation (or features) on which they are applied [39]. Constructing efficient and accurate representations for high-dimensional, complex MS vectors is a fundamental challenge in the field of MS data analysis. Particularly when using raw MS data for deep learning classification, how to effectively transform raw, high-dimensional MS vectors into an input representation that can be efficiently learned by models is a key problem worthy of exploring.

This study proposes a supervised representation learning module based on multi-channel embedding, MSMCE, which converts raw MS vectors into multi-channel embedded representations, significantly enhancing feature expressiveness, thereby optimizing the input for downstream deep learning models. Results show that this method also achieves significant performance improvements on datasets with numerous classes and limited sample sizes. Compared to traditional single-channel representations, the multi-channel embedding representation demonstrates significant advantages in improving classification accuracy, model training stability, and generalization capability. This is validated in Tables 2–4, where the Baseline Transformer exhibits extremely poor classification performance when directly processing high-dimensional, single-channel data. The model falls into what is known in machine learning as "mode collapse", a phenomenon where it converges to predict a single class for all samples. In contrast, the MSMCE module effectively avoids this issue by providing a more richly expressive and structured input, ensuring that the model can converge properly and learn discriminative feature representations. Additionally, ablation studies further validate the important contributions of key components within the MSMCE module, including Encoder, Channel Embedding, and Channel Concatenation, to enhancing feature representation and classification performance. Therefore, the proposed MSMCE module provides a novel and efficient feature representation method for deep learning classification directly using raw MS data, effectively addressing

the limitations of representational power inherent in single-channel representations, while also expanding the application prospects of representation learning in the field of MS data analysis.

### Limitations and future directions

Although this study has made significant progress, several aspects require further exploration. First, while multi-channel embedding demonstrates robustness on multi-class, small-sample datasets, its performance may still be limited in extreme cases, such as severely imbalanced class distributions. This is because deep learning models rely heavily on data, and when the sample size is too small, the model may fail to effectively learn the feature representations in the embedding space, leading to training failure. Second, the choice of embedding dimensions and parameter optimization still require dataset-specific tuning. Future research could incorporate adaptive feature selection mechanisms to make feature representations more dynamic and better suited for different datasets. Lastly, a major limitation of deep learning models in clinical decision-making is the lack of well-defined interpretability methods [40]. While multi-channel embedding provides rich feature representations, a notable limitation of the current study is that we have not deeply investigated the direct biological interpretation of these learned features or how they correlate with specific clinically relevant molecular patterns. Our primary focus was on computational methodology and the empirical demonstration of performance gains achieved by the MSMCE module. Uncovering the biological significance of the features learned by the different channels within MSMCE embeddings—for example, by identifying which m/z regions or patterns contribute most to the classifications and correlating these with known biomarkers or biological pathways—is a complex task that would require further dedicated bioinformatics analysis and potentially experimental validation, extending beyond the scope of this initial methodological work. This remains an important open research question and a key direction for future studies aimed at enhancing the translational potential and interpretability of the MSMCE module in biomedical applications.

Future work can focus on improving the interpretability of the MSMCE module, further uncovering the biological significance of embedded features across different channels to enhance the credibility of the model in clinical applications. Additionally, future studies could evaluate the effectiveness of this module in transfer learning scenarios, exploring its transferability across different datasets and tasks to assess its generalizability in mass spectrometry data analysis.

### Conclusion

This study proposes a supervised representation learning module based on multi-channel embedding, MSMCE, which transforms raw MS vectors into multi-channel embedded representations, significantly enhancing feature expressiveness. Experimental results demonstrate that this method not only improves classification accuracy but also enhances model stability and generalization capability. Additionally, ablation studies validate the critical roles of the Encoder, Channel Embedding, and Channel Concatenation, further proving the effectiveness of the MSMCE module in feature learning. Overall, this study provides a novel feature representation method for raw MS data analysis, expanding the application prospects of representation learning in the field and offering new technical support for raw MS data analysis.

### Code availability

The source code for this study is available on GitHub at https://github.com/WoodFY/MSMCE.

### Supporting information

**S1 File.  Estimated model size & FLOPs.** Tabulated summary of estimated peak GPU memory footprint (MB) and Floating-Point Operations (FLOPs) for baseline and MSMCE-enhanced deep learning models.
(XLSX)

**S2 File. Experiments Bootstrap confidence intervals.** Bootstrap 95% confidence intervals for mean performance metrics (Accuracy, Precision, Recall, F1-Score) from k-fold cross-validation on the hold-out test set for all evaluated models. (XLSX)

## Acknowledgment

This work was received computational support from the National Institute of Metrology, China.

## Author contributions

**Conceptualization:** Xingchuang Xiong.

**Data curation:** Yinchu Wang, Lin Guo, Wei Zhang.

**Investigation:** Fengyi Zhang, Yinchu Wang, Lin Guo, Wei Zhang.

**Methodology:** Fengyi Zhang.

**Project administration:** Xingchuang Xiong.

**Resources:** Boyong Gao, Xingchuang Xiong.

**Software:** Fengyi Zhang.

**Supervision:** Boyong Gao, Xingchuang Xiong.

**Validation:** Fengyi Zhang.

**Visualization:** Fengyi Zhang.

**Writing – original draft:** Fengyi Zhang.

**Writing – review & editing:** Xingchuang Xiong.

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
