## [Decision Letter · Decision Letter 0]

6 May 2025

PONE-D-25-08088MSMCE: A Novel Representation Module for Classification of Raw Mass Spectrometry DataPLOS ONE

Dear Dr. Xiong,

Thank you for submitting your manuscript to PLOS ONE. After careful consideration, we feel that it has merit but does not fully meet PLOS ONE’s publication criteria as it currently stands. Therefore, we invite you to submit a revised version of the manuscript that addresses the points raised during the review process. Please submit your revised manuscript by Jun 20 2025 11:59PM. If you will need more time than this to complete your revisions, please reply to this message or contact the journal office at plosone@plos.org . Please include the following items when submitting your revised manuscript:

We look forward to receiving your revised manuscript.

Kind regards,

Hirenkumar Kantilal Mewada

Academic Editor

PLOS ONE

Journal Requirements:

Reviewers' comments:

Reviewer's Responses to Questions

**Comments to the Author**

1. Is the manuscript technically sound, and do the data support the conclusions?

Reviewer #1: Yes

Reviewer #2: Partly

Reviewer #3: Yes

2. Has the statistical analysis been performed appropriately and rigorously? 

Reviewer #1: No

Reviewer #2: No

Reviewer #3: I Don't Know

3. Have the authors made all data underlying the findings in their manuscript fully available?

Reviewer #1: Yes

Reviewer #2: Yes

Reviewer #3: Yes

4. Is the manuscript presented in an intelligible fashion and written in standard English?

Reviewer #1: Yes

Reviewer #2: Yes

Reviewer #3: Yes

5. Review Comments to the Author

Reviewer #1: This manuscript presents MSMCE, a novel deep learning-based feature representation module designed for the classification of raw mass spectrometry (MS) data. The proposed method addresses known limitations of single-channel feature representations by introducing multi-channel embeddings with a residual-like concatenation strategy. The model is well-motivated, technically sound, and its performance is evaluated across four diverse and publicly available datasets using multiple neural network architectures (e.g., ResNet, DenseNet, LSTM, Transformer). The availability of code and data enhances the reproducibility and transparency of the study.

The authors demonstrate notable empirical improvements in classification accuracy and computational efficiency, supported by clear ablation studies. However, the manuscript would benefit from several key improvements to strengthen its statistical and practical contributions:

Major concerns:

1. While performance metrics (accuracy, precision, recall, F1) are thoroughly reported, the manuscript lacks statistical testing to validate the significance of the observed improvements. Repeated runs with different random seeds and the use of standard tests (e.g., Wilcoxon signed-rank test, bootstrapping confidence intervals) are recommended. This is particularly important for small-sample settings (e.g., Canine Sarcoma dataset).

2. The MSMCE module's effectiveness is well demonstrated quantitatively, but the biological meaning of the learned features is not explored. Given the biomedical context of mass spectrometry data, the manuscript would benefit from a discussion (or illustrative example) of how MSMCE embeddings relate to biologically meaningful features or clinically relevant patterns. This is important for building trust and enabling adoption in translational research.

3. The manuscript refers to a “residual connection” via channel concatenation, which is misleading, as no additive operation is applied. Please revise the terminology to avoid confusion with standard residual connections.

4. Details about hyperparameter selection, learning rate tuning, and batch sizes should be more explicitly stated or summarized in a table.

5. The methods for computing FLOPs and model size should be clearly described.

6. The t-SNE plots and radar charts in the supplementary material are helpful but not discussed in the main text. Please integrate interpretations of these visualizations into the Results or Discussion to show how MSMCE improves feature separability and efficiency.

Minor concerns:

• The manuscript is well-written in general. Minor edits are suggested for typographical consistency (e.g., consistent table formatting, figure references). Also, be precise with terms such as "residual connection."

• The references are not cited in sequential order (e.g., the citation sequence jumps from [3–8] to [10], or from [24–26] to [22,32,39]), which may confuse readers and should be corrected for consistency.

• Although ReLU and Dropout are standard components in deep learning, they should be briefly explained for clarity, given that PLOS ONE targets a multidisciplinary audience that may not be familiar with these concepts.

• Figures 1 to 3 are of very poor quality, with unreadable text. They should be replaced with higher-resolution versions to ensure legibility and clarity.

Reviewer #2: Authors have proposed a novel perspective and strategy for using feature-representations from convolutional neural network layers in analyzing complex, and spatially and temporally intertwined MS data. In the proposed MSMCE, the feature-representations from two CNN layer (referred to as channel) has been used to enhance data representation. This multi –channel representation, accompanying an encoder layer and a feature integration strategy has enhanced performance of the model and decreased computational cost of model training. Authors have compared their proposed model to existing models and have done an ablation study to show contribution of embedded channels.

While the novel innovation and proposed technique enhances the performance of the model and opens a new perspective, the paper lacks major details and statistics necessary for scientific reporting. The most important missing aspect is proper randomization and testing of the model. It is not clear if the reported enhancement is significant in comparison to the existing models. From the ablation study (table 7), I am convinced that the multi-channel embedding improves the performance, but I have major concerns about how the results have been reported. A lot of details about the models, their hyperparameters, training and testing , and regularization strategies are missing, making it hard to properly evaluate and compare contribution for multi-channel embedding.

Here are a few notes:

The last sentence of the abstract: “Experimental results ...MS data classification.” The link between reduced computational resource and generalizability of the model is not given and trivial.

This sentence in the introduction: “The large ... cancer detection”. The challenges need to be explicitly included. Explicitly including the challenges help to develop motivation of the paper.

This sentence: “These operations ... MS signal acquisition,” the issues need to be explicitly listed: such as peak shift, etc

This sentence: “Nevertheless, the high dimensionality of MS data combined with limited sample sizes makes it challenging for traditional methods to effectively meet the demands of data analysis [11].”, It is not the high dimensionality of data or small sample size that limits the performance of the traditional methods, but the mismatches between bathes and the need for preprocessing of data so the methods work. Actually, deep learning models need more data to be trained on compared to conventional methods especially if the data has higher complexity and dimensions. Reference 11 is not correctly referenced as it talks about using matrix factorization and using Bayesian framework to overcome such problems and does not explicitly relay the logic used in this sentence. Regardless, the sentence is not following logic and message of this paragraph.

This sentence: “These methods often fail to fully exploit the latent information within the data, thereby limiting their overall performance.”. No reference. This claim is not accurate and somewhat controversial. The concern is: these methods (if used correctly) are robust as cited in the same paragraph, but preprocessing data to feed to these models require careful fine tunings. Needs to be rewritten, maybe restructuring like: Achieving optimal performance with these methods typically involves multiple stages of preprocessing and fine-tuning of model parameters to fully leverage the rich, latent information embedded in complex MS data.

This sentence:” In recent years, deep learning has emerged as a dominant technology”. Deep learning is not a technology, rather an approach or methodology.

The term channel in this article has been used in title and as a cornerstone of the article. For the first-time reader, the term is confusing, because, in the context of MS data, channel does not have a single universal definition. The authors are using channel as a concept of feature representations from CNN layers. This becomes clear later in the methods section. The term channel needs to be clarified and defined explicitly in the introduction. There is also some mix-up of term channel and concept of dimension. For example, in this sentence: “In the field of image classification, multi-channel images contain complementary information across different channels, enabling a more comprehensive description of the target compared to single-channel representations”. Consider confusion in this sentence: “These multi-channel features not only enrich feature representations but also enhance the model’s adaptability to high-dimensional data.” I suggest definition of term “channel” in the introduction, and then being careful not to use it in interchange with dimension- representation- feature-etc

This sentence does not read well: MS2DeepScore [25] employs a Siamese neural network to learn low-dimensional embeddings of MS vectors for predicting the structural similarity between chemical compounds, which is also applicable to MS Clustering. Maybe a comma is missing after vectors?

In this sentence: Studies have shown that deep learning can directly capture complex patterns”, what is the implication of the word “directly”? What would the indirect way be? I suggest removing the word “directly”

deep classification model” is not a valid term.

In the section where authors outline the contribution of this study:

“End-to-End Training Framework”, training the DL+classifier model in an end-to-end manner is not innovative and is widely used. See: Seddiki, Khawla, et al. "Early diagnosis: End-to-end CNN–LSTM models for mass spectrometry data classification." Analytical Chemistry 95.36 (2023): 13431-13437.,

“Dimensional Adaptation of Multi-Channel Embedding Representations:” This contribution is not clear. It is not clear what authors mean by “adaptability”. “Compared to the original single-channel MS vectors, the embedded multi-channel vectors exhibit better adaptability within CNN architectures.”, how has the adaptability been quantified? What are the statistics of the enhanced adaptability, where is the pvalue?

A general feedback to the authors. Authors have used a fully connected layer as the first layer. It is not wrong and authors’ choice to do so, but in my opinion, it is counter intuitive to use a flat layer to reduce dimensionality. The local MS information is lost this way. When transforming using WX+b, all local dependence is lost (for example peak shape, etc). This is counter-intuitive with using the CNN in the rest of the structure of the model. In most of the cited papers, usually a CNN layers comes first ensuring capturing the local dependencies of peaks, etc. But again, the proposed model by the authors is valid.

E is not a “feature matrix” in the classic sense (i.e., not raw features you designed or extracted). It is more accurate to call it: A latent embedding, or A learned representation

Figure 1: input is BxD but in the figure it is illustrated as 1-D

Addition of channel dimension is poorly shown in the figure 1 making it harder to understand. The green input vector (which says embeding_dim) directly goes into 1D convolution skipping adding a dimension to.

The convolutions are 1D, and the Eprime is Bx1xd, what is “3” in dimension of K1 (1xC/2x3), which dimension the convolution is happening? The same question for K2. It seems the notion and the mapping between the size of K1 and K2 and the output is inconsistent.

“By transforming feature vectors into multi-channel representations, this module enhances adaptability for downstream classification tasks.” this claim needs to be proven in a quantifiable way. It is not clear what does adaptability mean.

The training procedure depicted in figure 2 is in contrast with what has been mentioned as “End-to-End Training Framework”, Figure 2 is showing the backpropagation path back to only classifier module leaving out the MCMCE.

“enabling an adaptive representation of the input.” the term “adaptive” has not been used correctly.

2.6 Data Processing Workflow: This section starts with a few paragraphs as its own introduction. These paragraphs need to be moved to the introduction or discussion depending on the context and in this section only the method should be discussed.

The data processing section leaves most of the details out, referencing [2],[10]. details of the method should be outlined.

Section 2.7- What is the strategy to divide data to training, validation, and test sets? Has k-fold strategy been used? How many times the model has been trained? What is the strategy to avoid overfitting? Are all the results related to a single (but same for all models) trial division? Or the random seed might be different for each tested model?

In the table 2,3 and 5 there are values for precision and F1 that are below 0.5. This does not make sense, because any model that has been trained on data should be better than a totally random model with 50% performance. Also, a value of exactly 0.5000 for recall seems not to be correct considering variability in the MS data.

Table 2 and 3 does not report number of trained models or any variability related to the performance. The absolute value shows enhancement when the MSMCE has been included. However, no significance statistical metric has been reported. The proper way would have been multiple trainings using multiple random divisions of trials and then reporting an average, std and a pvalue for each comparison.

“representing an increase of 1.608%”, this enhancement is not of value, unless repeated over several randomizations and then averaged.

Tables 2, 3,5 and 6 report performance for models LSTM and transformer, without providing details of architecture of each of these models. LSTM layers or transformer architecture could be used in a DL-based pipeline in many ways. Details need to be properly reported.

Table 6 (class12), precision and recall reported for multi-class arrangement. It is not clear if the final precisions the average of precisions for each class or a different strategy has been used.

Figure 4, the reported accuracy drops at some epochs, which makes the reported accuracy metric to be questionless considering optimizers like Adam and loss functions like cross-entropy. How do the authors justify the drops?

ResNet-50, DenseNet-121, and EfficientNet-B0 should be properly referenced.

Figure 5, the lines should be transparent (alpha=0.8) for better visibility. The lines are blocking each other.

Reviewer #3: I would like to begin by congratulating the authors for this very interesting and well-executed piece of work.

This well-written article introduces a novel spectral representation technique that replaces traditional pre-processing methods with a neural network model employing multi-channel embedding.

The model is clearly presented and supported by a well-structured introduction. The limitations of the approach are adequately discussed in both the Materials and Methods section and the Discussion. However, I have several suggestions to enhance the clarity and completeness of the manuscript:

Image Quality: The figures are currently of insufficient quality, making them unreadable. This significantly hinders the reader's ability to follow the results. I strongly recommend improving the resolution and clarity of all figures to ensure they are legible and informative.

Description of the Data: More detailed information about the dataset is necessary. Readers unfamiliar with this data may not understand its specific challenges, particularly in the context of classification tasks. Please elaborate on the nature of the dataset, potential difficulties, and how these may relate to the performance of the proposed representation technique.

Error and Robustness Metrics: While the provided metrics illustrate the model’s performance, the lack of error analysis is a limitation. Including confidence intervals or other measures of uncertainty would provide a clearer picture of the method’s robustness. It would also facilitate a more rigorous comparison between methods.

Comparison with Traditional Methods: The proposed technique is positioned as an improvement over traditional feature engineering methods. However, the manuscript does not clearly specify which traditional techniques were used to generate the “Original” baseline. Please provide more details about these baseline methods so readers can better assess the added value of MSMCE.

Once again, I thank the authors for their contribution, and I also thank the editors for giving me the opportunity to review this manuscript.

6. PLOS authors have the option to publish the peer review history of their article (what does this mean? ). If published, this will include your full peer review and any attached files.

**Do you want your identity to be public for this peer review?** For information about this choice, including consent withdrawal, please see our Privacy Policy .

Reviewer #1: No

Reviewer #2: **Yes: ** Amir Akbarian

Reviewer #3: No

---

## [Author Response · Author response to Decision Letter 1]

27 May 2025

We have uploaded the Response to Reviewers.docx. All responses are in this file.

Reviewer #1

<COMMENT #1-1> 1. While performance metrics (accuracy, precision, recall, F1) are thoroughly reported, the manuscript lacks statistical testing to validate the significance of the observed improvements. Repeated runs with different random seeds and the use of standard tests (e.g., Wilcoxon signed-rank test, bootstrapping confidence intervals) are recommended. This is particularly important for small-sample settings (e.g., Canine Sarcoma dataset).

<RESPONSE #1-1> Thank you for this important suggestion regarding statistical validation of our results. We have addressed this by incorporating more rigorous statistical analysis in our revised manuscript.

Specifically:

Our experimental design, as detailed in Section 'Experimental Setup and Training Strategy', already employs a k-fold stratified cross-validation (with k=6]) where each model is trained k times on different folds of the training data and evaluated on a common hold-out test set. This provides multiple performance measurements for each model on the test set.

Bootstrap Confidence Intervals: We now report the 95% bootstrap confidence intervals for the mean performance metrics (Accuracy, Precision, Recall, F1-score) obtained from these k-fold evaluations on the hold-out test set. These confidence intervals provide a measure of the uncertainty around our reported mean values and are summarized in the table provided in the Supporting Information (Supporting File S3).

Wilcoxon Signed-Rank Test for Paired Comparisons: To assess the statistical significance of the improvements observed when using our MSMCE module compared to baseline models (i.e., models without MSMCE), we have performed paired Wilcoxon signed-rank tests. These tests compare the performance metrics (e.g., accuracy) of the MSMCE-enhanced model versus its corresponding baseline model across the k test set evaluations from the k-fold cross-validation. The resulting p-values are now reported in the text when discussing these comparisons in the 'Experiment and Results' section (e.g., "MSMCE-ResNet50 showed a statistically significant improvement in accuracy over the baseline ResNet-50 (p = 0.03125)").

While we used a fixed global random seed for reproducibility in the current k-fold cross-validation framework, the use of k-fold itself provides multiple evaluation points. We believe these additions, particularly the inclusion of bootstrap confidence intervals and Wilcoxon signed-rank test results, now provide the necessary statistical validation for the observed performance improvements, especially for datasets like Canine Sarcoma.

<COMMENT #1-2> 2. The MSMCE module's effectiveness is well demonstrated quantitatively, but the biological meaning of the learned features is not explored. Given the biomedical context of mass spectrometry data, the manuscript would benefit from a discussion (or illustrative example) of how MSMCE embeddings relate to biologically meaningful features or clinically relevant patterns. This is important for building trust and enabling adoption in translational research.

<RESPONSE #1-2> We sincerely thank the reviewer for this insightful comment and for highlighting the importance of exploring the biological meaning of the learned features, especially given the biomedical context of mass spectrometry data. We agree that understanding how MSMCE embeddings might relate to biologically significant features or clinically relevant patterns is crucial for building trust and facilitating translational research.

In the current study, our primary objective was to investigate, from a computational and data science perspective, a novel feature representation methodology for raw mass spectrometry data. Specifically, we focused on the development and empirical validation of the MSMCE and its impact on the classification performance and training efficiency of deep learning models. Our research aimed to determine whether transforming single-channel MS data into a multi-channel representation could provide tangible benefits for model training and predictive accuracy, as demonstrated by our quantitative results across four public datasets.

Therefore, while our current work establishes the computational advantages of the MSMCE module, we consider the interpretation of its learned features in a biological or clinical context as an important avenue for future research. We have now explicitly addressed this point in the revised manuscript by expanding the 'Limitations and future directions' subsection within the 'Discussion' section. In this revised section, we clearly state that uncovering the biological significance of the MSMCE embeddings is a key area for subsequent investigation.

We appreciate the reviewer's guidance in emphasizing this aspect, as it will undoubtedly shape our future work in this domain.

<COMMENT #1-3> 3. The manuscript refers to a “residual connection” via channel concatenation, which is misleading, as no additive operation is applied. Please revise the terminology to avoid confusion with standard residual connections.

<RESPONSE #1-3> We sincerely thank the reviewer for this precise and important observation. We acknowledge that our use of the term 'residual connection' in the context of channel concatenation could be misleading, as it does not involve the additive operation characteristic of standard residual connections found in architectures like ResNet. Our intention was to describe an operation that similarly aims to preserve and pass through earlier-stage features (the globally encoded feature E^') alongside newly transformed features (the multi-channel convolutional embeddings C_2).

To address this and ensure clarity, we have revised the manuscript to more accurately describe this operation. Specifically:

We have removed the term 'residual connection' where it might cause confusion with additive skip connections.

We now describe the operation explicitly as 'feature concatenation along the channel dimension to integrate global and local embeddings.'

The purpose of this concatenation—to combine the initial globally encoded representation E^' (with shape B×1×d) with the locally refined multi-channel convolutional features C_2 (with shape B×C×D) to form a richer, combined representation O (with shape B×(1+C)×d)—is now stated more directly without relying on the 'residual' analogy.

These changes have been implemented in the Abstract, Introduction, the 'Materials and Methods' section (specifically the subsection previously titled 'Channel “Residual” Connection', which has been renamed, and in its description), and the Discussion. We believe these revisions accurately reflect our methodology and prevent any potential misunderstanding with standard additive residual connections.

<COMMENT #1-4> 4. Details about hyperparameter selection, learning rate tuning, and batch sizes should be more explicitly stated or summarized in a table.

<RESPONSE #1-4> Thank you for this comment. We have ensured that details regarding hyperparameter selection and learning rate tuning are explicitly stated within the revised 'Experimental Setup and Training Strategy' section of the revised manuscript. Furthermore, the batch sizes used for the experiments are introduced in the preliminary description of the 'Experiment and Results' section (prior to detailing the dataset-specific results, of the revised manuscript). We believe these sections now provide the necessary clarity on these parameters.

<COMMENT #1-5> 5. The methods for computing FLOPs and model size should be clearly described.

<RESPONSE #1-5> Thank you for requesting clarification on the methods used to compute FLOPs and model size. We have added these details to the 'Computational Efficiency Analysis' subsection within the 'Experiment and Results' section of our revised manuscript.

Specifically:

FLOPs (Floating Point Operations): FLOPs were calculated using the thop library. This calculation was performed by providing a single sample tensor with dimensions (1, spectrum_dim) (where spectrum_dim is the length of the binned mass spectrum before MSMCE processing) as input to the profile function. FLOPs are reported in GigaFLOPs (GFLOPs).

Model Size (Number of Parameters): The 'model size' reported in our study refers to the Estimated Total Size (MB) of GPU memory occupied by the model during one forward and one backward pass. This was obtained using the torchinfo.summary function, with an input size corresponding to the batch size used during training (i.e., (batch_size, spectrum_dim)). This metric provides an estimate of the peak memory footprint during training.

<COMMENT #1-6> 6. The t-SNE plots and radar charts in the supplementary material are helpful but not discussed in the main text. Please integrate interpretations of these visualizations into the Results or Discussion to show how MSMCE improves feature separability and efficiency.

<RESPONSE #1-6> Thank you for this valuable suggestion. We agree that discussing these visualizations in the main text will enhance the manuscript.

t-SNE Visualizations: We have now incorporated a discussion of the t-SNE visualizations (provided in the supplementary material, illustrating the data distributions for each dataset) into the preliminary description of the 'Experiment and Results' section. This addition aims to provide an initial qualitative insight into the class separability of the datasets before the application of our models. While these t-SNE plots are based on the original processed data features (prior to MSMCE embeddings) and thus do not directly show how MSMCE improves feature separability, they offer a baseline understanding of the dataset’s characteristics.

Radar Charts: We would like to clarify that the radar charts, which illustrate model efficiency (FLOPs, model size) in relation to accuracy, are already discussed in detail within the 'Computational Efficiency Analysis' subsection of the 'Experiment and Results' section. This discussion explicitly highlights how MSMCE improves computational efficiency.

<COMMENT #1-7> 7. The manuscript is well-written in general. Minor edits are suggested for typographical consistency (e.g., consistent table formatting, figure references). Also, be precise with terms such as "residual connection".

<RESPONSE #1-7> We appreciate the reviewer's positive feedback on the overall writing. We have carefully reviewed the entire manuscript and made edits to ensure typographical consistency, including table formatting and figure references. Furthermore, we have taken care to use terminology precisely. For instance, as per earlier feedback and the reviewer's note, the term analogous to 'residual connection' which is achieved via channel concatenation has been revised to more accurately describe the operation as 'channel concatenation', avoiding any misleading association with standard additive residual connections.

<COMMENT #1-8> 8. The references are not cited in sequential order (e.g., the citation sequence jumps from [3–8] to [10], or from [24–26] to [22,32,39]), which may confuse readers and should be corrected for consistency.

<RESPONSE #1-8> Thank you for pointing out the inconsistencies in our citation order. We have thoroughly reviewed and revised the manuscript to ensure that all references are now cited in strict numerical sequence according to their first appearance in the text, and that the reference list is ordered accordingly. This has been corrected throughout the manuscript for consistency.

<COMMENT #1-9> 9. Although ReLU and Dropout are standard components in deep learning, they should be briefly explained for clarity, given that PLOS ONE targets a multidisciplinary audience that may not be familiar with these concepts.

<RESPONSE #1-9> We thank the reviewer for this thoughtful suggestion. We agree that providing brief explanations for standard deep learning components like ReLU and Dropout is beneficial for enhancing the clarity and accessibility of our manuscript to a multidisciplinary readership, which PLOS ONE serves.

In the revised manuscript, we have incorporated concise definitions and the purposes of both ReLU and Dropout. These explanations have been added to the 'Materials and Methods' section, within the subsection 'Fully Connected Encoder', where these components are first introduced as part of our MSMCE module's architecture. We believe this will help readers who may not be deeply familiar with these specific deep learning techniques to better understand their roles within our proposed model. Specifically, we have clarified that:

ReLU (Rectified Linear Unit) is an activation function that introduces non-linearity into the model, outputting the input directly if it is positive, and zero otherwise, which helps with issues like vanishing gradients and computational efficiency.

Dropout is a regularization technique used during training where a proportion of neuron outputs in a layer are randomly ignored, which helps prevent overfitting and improves the model's generalization to unseen data.

<COMMENT #1-10> 10. Figures 1 to 3 are of very poor quality, with unreadable text. They should be replaced with higher-resolution versions to ensure legibility and clarity.

<RESPONSE #1-10> We apologize for the issues with the figure quality experienced by the reviewer. We believe this may be due to the compression of images when the journal generates the review manuscript. We had initially submitted high-resolution figures. For this resubmission, we have re-checked and ensured that Fig 1 to 3 (and all other figures) are provided in high resolution, meeting PLOS ONE’s image guidelines, to ensure their legibility and clarity. We have also utilized the PACE tool as recommended to verify figure compliance.

Reviewer #2

<COMMENT #2-1> 1. The last sentence of the abstract: “Experimental results ...MS data classification.” The link between reduced computational resource and generalizability of the model is not given and trivial.

<RESPONSE #2-1> We thank the reviewer for pointing out the need for greater clarity regarding the link between reduced computational resources/enhanced training efficiency and model generalizability in the abstract. We agree that this connection is not direct or self-evident and that our original phrasing might have inadvertently implied a direct causal relationship.

Our intention was to highlight that while the primary contributions of MSMCE are significant improvements in classification performance and computational efficiency, the latter (enhanced efficiency) can indirectly facilitate the development of more generalizable models. For instance, reduced training time and lower resource consumption allow for more extensive hyperparameter tuning, cross-validation, and experimentation with different model architectures or larger datasets, all of which are crucial for building models that generalize better to unseen data.

However, to avoid any misinterpretation and to maintain a precise focus on the direct, demonstrated benefits, we have revised the concluding part of the abstract. The revised sentence now emphasizes the demonstrated effectiveness in classification and efficiency, and more cautiously suggests its potential for broader applicability, without making a strong, unproven claim about directly enhancing generalizability due to resource reduction.

The revised sentence in the abstract now reads: ' Experimental results on four public datasets demonstrate that the proposed MSMCE module not only achieves substantial improvements in classification performance but also enhances computational efficiency and training stability, highlighting its effectiveness in raw MS data classification and its potential for robust application across diverse datasets.'

<LOCATION #2-1> Abstract section, final sentence.

<COMMENT #2-2> 2. This sentence in the introduction: “The large ... cancer detection”. The challenges need to be explicitly included. Explicitly including the challenges help to develop motivation of the paper.

<RESPONSE #2-2> We thank the reviewer for this valuable suggestion. We agree that explicitly listing the challenges associated with MS data analysis in the introduction will better establish motivation for our

---

## [Decision Letter · Decision Letter 1]

19 Jun 2025

PONE-D-25-08088R1MSMCE: A Novel Representation Module for Classification of Raw Mass Spectrometry DataPLOS ONE

Dear Dr. Xiong,

Thank you for submitting your manuscript to PLOS ONE. After careful consideration, we feel that it has merit but does not fully meet PLOS ONE’s publication criteria as it currently stands. Therefore, we invite you to submit a revised version of the manuscript that addresses the points raised during the review process.

We look forward to receiving your revised manuscript.

Kind regards,

Hirenkumar Kantilal Mewada

Academic Editor

PLOS ONE

Reviewers' comments:

Reviewer's Responses to Questions

**Comments to the Author**

1. If the authors have adequately addressed your comments raised in a previous round of review and you feel that this manuscript is now acceptable for publication, you may indicate that here to bypass the “Comments to the Author” section, enter your conflict of interest statement in the “Confidential to Editor” section, and submit your "Accept" recommendation.

Reviewer #1: All comments have been addressed

Reviewer #2: (No Response)

Reviewer #3: All comments have been addressed

2. Is the manuscript technically sound, and do the data support the conclusions?

Reviewer #1: Yes

Reviewer #2: Yes

Reviewer #3: Yes

3. Has the statistical analysis been performed appropriately and rigorously? 

Reviewer #1: Yes

Reviewer #2: No

Reviewer #3: Yes

4. Have the authors made all data underlying the findings in their manuscript fully available?

Reviewer #1: Yes

Reviewer #2: Yes

Reviewer #3: Yes

5. Is the manuscript presented in an intelligible fashion and written in standard English?

Reviewer #1: Yes

Reviewer #2: Yes

Reviewer #3: Yes

6. Review Comments to the Author

Reviewer #1: The authors have provided a comprehensive and thoughtful revision. All reviewer concerns were addressed appropriately and with substantial improvements to both the clarity and rigor of the manuscript. Key contributions such as the statistical validation of performance improvements, clarification of convolutional architecture and terminology, and enhanced methodological detail, strengthen the manuscript considerably.

Reviewer #2: The reported p-values appear inconsistent and potentially incorrect. All five instances report the exact same value (p = 0.03125), regardless of the magnitude of performance improvement. For example, in Table 5, the accuracy of the Transformer model on the Canine Sarcoma dataset increases markedly from 0.78 ± 0.00 to 0.99 ± 0.01 — a substantial improvement that would typically yield a much lower p-value (e.g., p < 0.01) if statistical significance were properly assessed. The uniformity of the reported p-values across different metrics and datasets raises concerns about the validity of the statistical testing procedure. Clarification is needed on how these p-values were computed and whether appropriate statistical tests were applied in each case.

As noted in the previous round, the performance values reported in Tables 2, 3, and 4 — such as 0.25, 0.33, and 0.4 — are notably lower than what would be expected from a random classifier in a binary classification task. This suggests that the number of model training iterations may have been insufficient to yield a reliable evaluation. The authors report using 6-fold cross-validation (k=6); however, this was applied to a single 90/10 train-test split, meaning the same data partitioning was used throughout the evaluation (i.e., n=1). Given the class imbalance evident in the t-SNE plots provided in the Supplementary Materials, repeating the 90/10 split multiple times is essential to ensure a fair comparison between the baseline and enhanced models. Each repetition can be followed by k-fold cross-validation (potentially with a lower k if computational feasibility is a concern). This approach helps mitigate the impact of any train-test split, smoothing out performance fluctuations caused by randomness in the data. For instance, a low score such as 0.25 from one split may be balanced by higher scores in others, leading to a more representative average.

The p-values in the tables appear to be reported selectively and not consistently across all model comparisons. Specifically, only three p-values are provided: the accuracy of ResNet-50 on the NSCLC dataset, the accuracy of DenseNet-121 on the CRLM dataset, and the F1-score of DenseNet-121 on the RCC dataset. Reporting statistical significance solely for accuracy — without corresponding significance measures for other relevant metrics such as precision or recall — provides an incomplete assessment of model performance. This is particularly concerning in imbalanced classification tasks, where improvements in accuracy may mask critical deficiencies, such as an increase in false positives or false negatives. To ensure scientific rigor and transparency, all key performance metrics reported in the tables — especially those being used to support claims of improvement — should be accompanied by appropriate statistical tests and corresponding p-values. Furthermore, if a reported result is not statistically significant, it should be clearly labeled as such or omitted to avoid misinterpretation.

Line 646: “FLOPs of all these models significantly decrease”, the p-value of the significance, the test name, and the sample size for the test are missing for this claim.

Minor issues:

In the supporting material, the dots on the figures need to be transparent and with a narrow margin for each marker. The points are obstructing each other.

The test for 1 p-value has been reported as signrank test but others are missing the statistical test name. Adding a sentence to clarify that the same test has been used to calculate all significance levels could help clarify. Please also include number of samples used to calculate p value (probably k=6).

Line 542-546, Lines 542–546 — The term “training crash” is not clearly defined or quantified, making it difficult to interpret its impact or relevance. I recommend moving lines 542–554 to the Discussion section, as this content is largely descriptive and does not report any measurable or validated performance metrics. Relocating this section would improve the logical structure of the Results section by keeping it focused on objective findings.

Line 547- Line 547 — The use of the term “excellent” to describe the model’s performance is not appropriate in the context of a scientific report. Descriptive terms such as this are subjective and should be replaced with objective, metric-based statements that accurately reflect the quantitative results.

Line 580, Thie seems to be a typo.

Line 433 Using the term Experimental is not appropriate for this section. Similarly line 489. The model fitting and testing has been performed in silico and no experiment was performed in a lab.

In figure 4, the dashed lines are reporting accuracy over “val”. However, the term “val” needs to be replaced with “test”, as no k-fold validation has been used for the ablation study, and the performance is measured over test set. Similarly, the term “validation performance” needs to be corrected as “performance over the test set”.

Line 644, abbreviation of term FLOP has been repeated twice.

Reviewer #3: I would like to thank the authors for their careful and comprehensive revisions in response to my previous comments. All of the concerns I raised have been thoroughly addressed, and the manuscript has been substantially improved in both clarity and scientific rigor.

I have two additional recommendations that, while not essential, may further enhance the quality and readability of the manuscript:

1. Inclusion of Supplementary Figures (A to C):

The t-SNE visualizations currently provided in the Supplementary Information (Figures A to C) offer valuable insights into the intrinsic structure and difficulty of the classification tasks associated with each dataset. If feasible, I recommend incorporating these figures into the main manuscript to strengthen the presentation of the dataset characteristics and facilitate the reader’s understanding of the challenges involved.

2. Presentation of Bootstrap Confidence Intervals:

Supplementary Table S3, which reports 95% bootstrap confidence intervals for the mean performance metrics, provides important information on the robustness of the results. I encourage the authors to consider including these intervals directly in the main performance tables of the manuscript (e.g., Tables 2 to 6), so that the uncertainty associated with each metric is readily visible to the reader.

These are minor suggestions aimed at improving the accessibility and completeness of the manuscript. I commend the authors for their significant efforts and support the publication of the revised version, with or without the incorporation of these final recommendations.

7. PLOS authors have the option to publish the peer review history of their article (what does this mean? ). If published, this will include your full peer review and any attached files.

**Do you want your identity to be public for this peer review?** For information about this choice, including consent withdrawal, please see our Privacy Policy .

Reviewer #1: **Yes: ** Raquel Cumeras

Reviewer #2: **Yes: ** Amir Akbarian

Reviewer #3: No

---

## [Author Response · Author response to Decision Letter 2]

26 Jun 2025

We have uploaded the Response to Reviewers.docx. All responses are in this file.

Thank you for your letter dated June 19, 2025, and for the opportunity to revise our manuscript titled "MSMCE: A Novel Representation Module for Classification of Raw Mass Spectrometry Data" (PONE-D-25-08088). We also extend our sincere gratitude to the reviewers for their insightful comments and constructive suggestions, which have been invaluable in improving the quality and clarity of our paper.

We have carefully considered all the points raised by the Academic Editor and the reviewers. We believe that the revisions made have substantially strengthened the manuscript. We have addressed each comment in a point-by-point manner in the file of "Response to Reviewers", and for ease of review, we have highlighted the revisions in bold. The changes in the revised manuscript have been marked using the "Track Changes" feature in Microsoft Word, as requested. We have also uploaded a clean, unmarked version of the revised manuscript.

We hope that the revised manuscript is now suitable for publication in PLOS ONE.

<COMMENT #2-1> 1. The reported p-values appear inconsistent and potentially incorrect. All five instances report the exact same value (p = 0.03125), regardless of the magnitude of performance improvement. For example, in Table 5, the accuracy of the Transformer model on the Canine Sarcoma dataset increases markedly from 0.78 ± 0.00 to 0.99 ± 0.01 — a substantial improvement that would typically yield a much lower p-value (e.g., p < 0.01) if statistical significance were properly assessed. The uniformity of the reported p-values across different metrics and datasets raises concerns about the validity of the statistical testing procedure. Clarification is needed on how these p-values were computed and whether appropriate statistical tests were applied in each case.

<RESPONSE #2-1> We sincerely thank the reviewer for their meticulous observation and for raising this important question regarding our p-value results. We completely understand why the recurrence of the exact same p-value (p = 0.03125) (We rounded 0.03125 to 0.03 in the current manuscript) across multiple instances would raise valid concerns about its validity. We would like to take this opportunity to clarify our statistical testing procedure. We confirm that all reported p-values were computed using the paired Wilcoxon signed-rank test on the performance scores from each of the 6 folds of our cross-validation (K=6). Therefore, the sample size for each test was N=6.

And most crucially, the reason for this identical p-value appears across instances with different magnitudes of performance improvement is a direct consequence of the properties of the Wilcoxon signed-rank test itself. As a non-parametric test, its calculation relies on the sign and ranks of the paired differences, not on the magnitude of these differences. In every instance where the p-value was reported as 0.03125, it was the case that our proposed MSMCE-enhanced model outperformed its corresponding baseline model in all 6 folds of the cross-validation. For a Wilcoxon signed-rank test with a sample size of N=6, the scenario where all 6 differences share the same sign (e.g., all positive) constitutes the most extreme and consistent positive result that the test can detect. In this "perfect sweep" situation, the test will always yield its minimum possible p-value, regardless of whether the performance improvement in each fold was marginal or substantial. For a two-sided test, this minimum p-value is precisely 0.03125. The detailed description is as follows:

Test Setup and Hypotheses

To clearly elaborate on our p-value calculation process, we detail here the specific steps of the paired Wilcoxon signed-rank test as applied in our case. As stated in the manuscript, we compare the paired performance scores (sample size N=6) of the "models integrated with MSMCE" against their "baseline" counterparts from the 6-fold cross-validation. The null hypothesis (H_0) for this test is that there is no difference in performance between the two models (i.e., the median of the performance differences is zero). Our two-sided test corresponds to the alternative hypothesis (H_a) that a difference in performance does exist (i.e., the median of the differences is not zero).

Calculating the Test Statistic (W)

The calculation begins by taking the difference d for each of the 6 pairs of performance scores. Subsequently, we rank the absolute values of these differences, |d|, from 1 to 6. The original signs (+ or -) are then reassigned to their corresponding ranks, and we separately calculate the sum of the positive ranks (W^+) and the sum of the negative ranks (W^-). Finally, the test statistic W is defined as the minimum of W^+ and W^-. The value of this W statistic reflects the consistency of the difference between the two groups.

Principle of the Two-Sided p-value Calculation

The p-value represents the probability of observing a W statistic as extreme as, or more extreme than (i.e., closer to 0), our calculated W, assuming the null hypothesis (no difference in model performance) is true. For a sample size of N=6, there are 2^6=64 possible combinations of signs for the ranks. Because we are conducting a two-sided test, we consider extreme outcomes in both directions: a very small W (indicating the MSMCE model is consistently better) and a very large W (indicating the baseline model is consistently better). The p-value is the sum of the probabilities of these two extreme scenarios.

Specific Calculation for p = 0.03125

The recurring p-value of 0.03125 in our manuscript corresponds to the most extreme result possible for this test, where the test statistic W=0. This scenario occurs only when one condition is met: the model integrated with MSMCE outperforms the baseline model in every one of the 6 folds. In this case, all 6 performance differences are positive, making the sum of negative ranks (W−) equal to 0, the probability of its occurrence is (1\/2)^6=1\/64, and therefore the test statistic W is also 0m, the probability of its occurrence is (1\/2)^6=1\/64. For this "perfect sweep" situation, the two-sided p-value is calculated as: (Probability of all differences being positive) + (Probability of all differences being negative) = (1/64) + (1/64) = 2/64 = 0.03125. This explains why this fixed, minimum p-value is obtained whenever a consistent improvement is observed, regardless of the magnitude of that improvement.

Therefore, the uniformity of the p-value is not an artifact of an incorrect procedure. On the contrary, it serves as strong evidence that the performance enhancement from our proposed MSMCE module is exceptionally robust and consistent across all data splits in those specific comparisons.

<COMMENT #2-2> 2. As noted in the previous round, the performance values reported in Tables 2, 3, and 4 — such as 0.25, 0.33, and 0.4 — are notably lower than what would be expected from a random classifier in a binary classification task. This suggests that the number of model training iterations may have been insufficient to yield a reliable evaluation. The authors report using 6-fold cross-validation (k=6); however, this was applied to a single 90/10 train-test split, meaning the same data partitioning was used throughout the evaluation (i.e., n=1). Given the class imbalance evident in the t-SNE plots provided in the Supplementary Materials, repeating the 90/10 split multiple times is essential to ensure a fair comparison between the baseline and enhanced models. Each repetition can be followed by k-fold cross-validation (potentially with a lower k if computational feasibility is a concern). This approach helps mitigate the impact of any train-test split, smoothing out performance fluctuations caused by randomness in the data. For instance, a low score such as 0.25 from one split may be balanced by higher scores in others, leading to a more representative average.

<RESPONSE #2-2> We sincerely thank the reviewer for this insightful and important comment. The concern regarding whether the low performance values (e.g., 0.25, 0.33, 0.4) were artifacts of a specific data split is entirely valid. A more robust validation protocol involving multiple independent splits is crucial for ensuring the reliability of our conclusions.

To that end, we have diligently followed your valuable recommendation and conducted a new, more comprehensive set of validation experiments. Specifically:

Multiple Independent Splits: We focused on the baseline Transformer model's performance on the NSCLC, CRLM, and RCC datasets. We repeated the 90/10 train-test split using three different random seeds (3407, 42, and 1234).

Nested Cross-Validation: For each of these independent splits, we subsequently performed a full 6-fold cross-validation on the corresponding training set.

NSCLC

Seed=3407 Fold Accuracy Precision Recall F1 Score

1 0.517062 0.258531 0.5 0.340831

2 0.482938 0.241469 0.5 0.325663

3 0.517062 0.258531 0.5 0.340831

4 0.482938 0.241469 0.5 0.325663

5 0.482938 0.241469 0.5 0.325663

Seed=42 Fold Accuracy Precision Recall F1 Score

1 0.490457 0.245228 0.5 0.329065

2 0.490457 0.245228 0.5 0.329065

3 0.509543 0.254772 0.5 0.337548

4 0.509543 0.254772 0.5 0.337548

5 0.490457 0.245228 0.5 0.329065

Seed=1234 Fold Accuracy Precision Recall F1 Score

1 0.484095 0.242047 0.5 0.326189

2 0.515905 0.257953 0.5 0.340328

3 0.484095 0.242047 0.5 0.326189

4 0.484095 0.242047 0.5 0.326189

5 0.484095 0.242047 0.5 0.326189

CRLM

Seed=3407 Fold Accuracy Precision Recall F1 Score

1 0.49542 0.24771 0.5 0.331292

2 0.49542 0.24771 0.5 0.331292

3 0.50458 0.25229 0.5 0.335362

4 0.50458 0.25229 0.5 0.335362

5 0.49542 0.24771 0.5 0.331292

Seed=42 Fold Accuracy Precision Recall F1 Score

1 0.500694 0.250347 0.5 0.333642

2 0.499306 0.249653 0.5 0.333025

3 0.500694 0.250347 0.5 0.333642

4 0.499306 0.249653 0.5 0.333025

5 0.500694 0.250347 0.5 0.333642

Seed=1234 Fold Accuracy Precision Recall F1 Score

1 0.494588 0.247294 0.5 0.330919

2 0.505412 0.252706 0.5 0.33573

3 0.505412 0.252706 0.5 0.33573

4 0.494588 0.247294 0.5 0.330919

5 0.505412 0.252706 0.5 0.33573

RCC

Seed=3407 Fold Accuracy Precision Recall F1 Score

1 0.314341 0.15717 0.5 0.239162

2 0.685659 0.34283 0.5 0.40676

3 0.685659 0.34283 0.5 0.40676

4 0.685659 0.34283 0.5 0.40676

5 0.685659 0.34283 0.5 0.40676

Seed=42 Fold Accuracy Precision Recall F1 Score

1 0.321383 0.160691 0.5 0.243217

2 0.678617 0.339309 0.5 0.404272

3 0.678617 0.339309 0.5 0.404272

4 0.678617 0.339309 0.5 0.404272

5 0.678617 0.339309 0.5 0.404272

Seed=1234 Fold Accuracy Precision Recall F1 Score

1 0.68694 0.34347 0.5 0.407211

2 0.68694 0.34347 0.5 0.407211

3 0.68694 0.34347 0.5 0.407211

4 0.68694 0.34347 0.5 0.407211

5 0.68694 0.34347 0.5 0.407211

The results from these new experiments (3 independent splits × 6 folds = 18 runs), are highly consistent with our original findings. This strongly indicates that the observed low scores are not an artifact of a specific data partition. Rather, they genuinely reflect the inherent challenge and insufficient representational power of the standard Transformer model when tasked with directly processing such high-dimensional and complex raw mass spectrometry data.

To further elucidate and validate this conclusion, we investigated the underlying reason for this poor performance. We hypothesized that these extremely low scores are due to the model entering a state of "Mode Collapse"—where it ceases to learn and instead predicts a single class for all samples. To provide direct evidence for this, we have now added confusion matrices for these low-performing baseline models to the Manuscript. These confusion matrices clearly visualize the model's extremely skewed predictions, offering intuitive proof that mode collapse has occurred.

In summary, we are very grateful for your suggestion. It has not only pushed us to strengthen the reliability of our study with more rigorous validation, but has also led us to provide a deeper and clearer explanation for the baseline models' performance bottleneck through the inclusion of confusion matrices.

<COMMENT #2-3> 3. The p-values in the tables appear to be reported selectively and not consistently across all model comparisons. Specifically, only three p-values are provided: the accuracy of ResNet-50 on the NSCLC dataset, the accuracy of DenseNet-121 on the CRLM dataset, and the F1-score of DenseNet-121 on the RCC dataset. Reporting statistical significance solely for accuracy — without corresponding significance measures for other relevant metrics such as precision or recall — provides an incomplete assessment of model performance. This is particularly concerning in imbalanced classification tasks, where improvements in accuracy may mask critical deficiencies, such as an increase in false positives or false negatives. To ensure scientific rigor and transparency, all key performance metrics reported in the tables — especially those being used to support claims of improvement — should be accompanied by appropriate statistical tests and corresponding p-values. Furthermore, if a reported result is not statistically significant, it should be clearly labeled as such or omitted to avoid misinterpretation.

<RESPONSE #2-3> We sincerely thank the reviewer for raising this extremely important point. We completely agree that comprehensive and consistent reporting of statistical test results is crucial for ensuring the rigor and transparency of our research, especially when dealing with imbalanced datasets. To systematically address the issues you have raised and to improve the clarity of our manuscript, we have made the following revisions:

Centralized Description of p-value Calculation: We have added a detailed paragraph describing our statistical analysis methodology in the introductory section of the "Results" section. This description clarifies that all p-values were uniformly calculated using a two-sided, paired Wilcoxon signed-rank test (N=6) on the results from our 6-fold cross-validation. This ensures the uniformity and transparency of our statistical methods.

Streamlined and Unified Table Content: To improve the readability and aesthetic presentation of the tables, while focusing on the most representative and comprehensive evaluation metrics, we have streamlined all performance comparison tables (Tables 2-6) in the manuscript. We have removed the display of Precision and Recall, retaining only Accuracy and F1-Score, which we believe are sufficient to reflect the overall performance of the models.

Comprehensive Inclusion of Key Metric p-values: In the streamlined tables, for every Accuracy and F1-Score corresponding to the MSMCE-integrated models, we have now included the p-value from the comparison against the respective baseline model. This ensures that all key performance metrics used to support our claims of improvement are accompanied by their corresponding statistical evidence.

Explanation for Non-significant Results: For any results where the performance improvement was found to be not statistically significant (p ≥ 0.05), we have now included explicit discussions and explanations in the main text of the manuscript. We discuss the potential reasons for this, such as the baseline model already reaching a performance ceiling, thereby avoiding any potential misinterpretation.

We believe that with these revisions—namely, a centralized methodological description, comprehensive p-value reporting for key metrics in tables, and discussion of non-significant results in the text—we have fully and rigorously addressed your concerns. Thank you again for your valuable feedback, which has greatly improved the quality of our manuscript.

<COMMENT #2-4> 4. Line 646: “FLOPs of all these models significantly decrease”, the p-value of the significance, the test name, and the sample size for the test are missing for this claim.

<RESPONSE #2-4> We thank the reviewer for this sharp observation. We would like to clarify that Floating Point Operations (FLOPs) is a deterministic metric, calculated from a fixed model architecture and input size. For any given model, this calculation yields a single,

---

## [Decision Letter · Decision Letter 2]

16 Jul 2025

MSMCE: A Novel Representation Module for Classification of Raw Mass Spectrometry Data

PONE-D-25-08088R2

Dear Dr. Xiong,

We’re pleased to inform you that your manuscript has been judged scientifically suitable for publication and will be formally accepted for publication once it meets all outstanding technical requirements.

Kind regards,

Hirenkumar Kantilal Mewada

Academic Editor

PLOS ONE

Additional Editor Comments (optional):

Reviewers' comments:

Reviewer's Responses to Questions

**Comments to the Author**

1. If the authors have adequately addressed your comments raised in a previous round of review and you feel that this manuscript is now acceptable for publication, you may indicate that here to bypass the “Comments to the Author” section, enter your conflict of interest statement in the “Confidential to Editor” section, and submit your "Accept" recommendation.

Reviewer #2: All comments have been addressed

Reviewer #3: All comments have been addressed

2. Is the manuscript technically sound, and do the data support the conclusions?

Reviewer #2: Yes

Reviewer #3: Yes

3. Has the statistical analysis been performed appropriately and rigorously? 

Reviewer #2: Yes

Reviewer #3: Yes

4. Have the authors made all data underlying the findings in their manuscript fully available?

Reviewer #2: Yes

Reviewer #3: Yes

5. Is the manuscript presented in an intelligible fashion and written in standard English?

Reviewer #2: Yes

Reviewer #3: Yes

6. Review Comments to the Author

Reviewer #2: All my previous concerns have been thoroughly resolved. The revised manuscript is significantly strengthened and presents a clear, well-supported, and insightful contribution to the field. I commend the authors for their rigorous approach, and the depth of their analysis. Congratulations on producing such a strong and impactful paper.

Reviewer #3: (No Response)

7. PLOS authors have the option to publish the peer review history of their article (what does this mean? ). If published, this will include your full peer review and any attached files.

**Do you want your identity to be public for this peer review?** For information about this choice, including consent withdrawal, please see our Privacy Policy .

Reviewer #2: **Yes: ** Amir Akbarian

Reviewer #3: No

---

## [Editor Report · Acceptance letter]

PONE-D-25-08088R2

PLOS ONE

Dear Dr. Xiong,

I'm pleased to inform you that your manuscript has been deemed suitable for publication in PLOS ONE. Congratulations! Your manuscript is now being handed over to our production team.

Kind regards,

on behalf of

Dr. Hirenkumar Kantilal Mewada

Academic Editor

PLOS ONE